# PHYSHANDI: PHYSICS-BASED RECONSTRUCTION OF HAND-DEFORMABLE OBJECT INTERACTIONS

## ABSTRACT

While existing methods for reconstructing hand–object interactions have made impressive progress, they either focus on rigid or part-wise rigid objects—limiting their ability to model real-world objects (e.g., cloth, stuffed animals) that exhibit highly non-rigid deformations—or model deformable objects without full 3D hand reconstruction. To bridge this gap, we present PHYSHANDI (**Phys**ics-based Reconstruction of **Hand** and **D**eformable Object **I**nteractions), a framework that enables full 3D reconstruction of both interacting hands and non-rigid objects. Our key idea is to *physically simulate* object deformations driven by forces induced from densely reconstructed 3D hand motions, ensuring that the reconstructed object dynamics are both physically plausible and coherent with the interacting hand movements. Furthermore, we demonstrate that such simulation of object deformations can, in turn, refine and improve hand reconstruction via inverse physics. In experiments, PHYSHANDI outperforms the state-of-the-art baseline across reconstruction, future prediction, and generalization to unseen interactions.

## 1 INTRODUCTION

The hand is our primary tool for interacting with objects, enabling a wide range of everyday object manipulation tasks (e.g., picking up a cell phone, folding clothes). Effective modeling of such hand–object interactions in 3D is crucial for enabling machines to perceive and reason about human actions, which in turn is important for applications such as immersive AR/VR experiences, robot learning from human demonstrations, and teleoperation. Owing to this importance, numerous studies have investigated the modeling hand–object interactions and reconstructing them from diverse sensing modalities, such as RGB images, depth maps, and RGB-D data (Hampali et al., 2020; Chao et al., 2021; Hasson et al., 2019; Mueller et al., 2017; Garcia-Hernando et al., 2018; Brahmbhatt et al., 2020; Taheri et al., 2020; Fan et al., 2023; Swamy et al., 2023; Corona et al., 2020; Damen et al., 2022; Brahmbhatt et al., 2019).

While these existing approaches have shown impressive progress, most of them are limited to modeling interactions with *rigid objects*. Although many real-world objects (e.g., cloth, charger cables) exhibit highly non-rigid deformation, most existing methods consider rigid or part-wise rigid objects in interaction (Hampali et al., 2020; Chao et al., 2021; Brahmbhatt et al., 2020; Taheri et al., 2020; Fan et al., 2023; Swamy et al., 2023; Corona et al., 2020; Damen et al., 2022; Brahmbhatt et al., 2019; Lee et al., 2024; Cho et al., 2024). Modeling and reconstructing such object dynamics is indeed straightforward, as they can be represented using a small set of rigid transformation matrices corresponding to each rigid body.

In contrast, non-rigid deformation involves complex, spatially varying dynamics with substantially higher degrees of freedom, making it harder to learn reliable dynamics from input data. While a few works have tackled hand–deformable object interaction modeling (Xie et al., 2023; Qi et al., 2025; Jiang et al., 2025), most of them (Xie et al., 2023; Qi et al., 2025) are limited to only small, localized deformations from *finger pressure* and do not readily extend to more general, large-scale non-rigid deformations. The most relevant work is PhysTwin (Jiang et al., 2025), which is capable of modeling large non-rigid deformations through *physical simulation*. Its focus, however, is primarily on reconstructing deformable objects *without full 3D hand reconstruction*. Instead, hands are represented by only a sparse set of points (whose cardinality is about 30) directly sampled from depth maps, which may limit the precision of interaction force modeling and lead to suboptimal model topology reconstruction for simulation, as discussed in later sections.

To address this, we introduce PHYSHANDI (**Phys**ics-based Reconstruction of **Hand** and **D**eformable Object **I**nteractions), a framework that enables *dense 3D reconstruction* of interacting hands and non-rigid objects through physics-based simulation. We represent hands using a dense parametric model (MANO model (Romero et al., 2017)) and objects using a classical physics-based model (Spring–Mass model (Liu et al., 2013; Jiang et al., 2025)) capable of simulating the dynamics of deformable objects. In particular, the simulation of object deformation is driven by forces induced from dense motions of MANO hand meshes, enabling the modeling of object dynamics that is both physically plausible and also coherent with the fully reconstructed interacting hand movements.

We propose an optimization pipeline to reconstruct this 3D dense hand–deformable object interaction model from sparse-view RGB-D videos. The pipeline consists of three stages: (1) hand reconstruction, (2) object reconstruction, and (3) hand refinement. In the *hand reconstruction* stage, we fit the MANO model (Romero et al., 2017) to the input RGB-D observations. In the *object reconstruction* stage, we fit the parameters of the Spring-Mass model (Liu et al., 2013; Jiang et al., 2025) conditioned on the reconstructed 3D hands. In particular, we simulate object deformations via spring–mass system (Liu et al., 2013; Jiang et al., 2025) driven by interaction forces induced from the reconstructed hand motions, and the resulting simulated object geometry is compared against the input RGB-D observations for parameter optimization. In the final *hand refinement* stage, we refine the initial hand reconstructions via inverse physics, leveraging the physics-based object model fitted in the previous stage. This refinement enforces that the reconstructed hands produce object simulations that are more consistent with the input observations. While we empirically find that the initial hand reconstruction stage is already sufficient to achieve state-of-the-art results with multi-view RGB-D inputs, this hand refinement stage proves especially effective when inference is performed from sparser inputs (e.g., future prediction in a single-view setting). To the best of our knowledge, this is the first work to demonstrate that inverse physics, guided by a physics-based deformable object model, can enhance hand reconstruction.

To experimentally validate the effectiveness of our method, we compare it against the most technically relevant baseline, PhysTwin (Jiang et al., 2025), a recent state-of-the-art approach for physics-based reconstruction of deformable objects in interaction with hands, and demonstrate that our method outperforms it in reconstruction, as well as future prediction and generalization to unseen interactions.

Our contributions can be summarized as follows:

- We present PHYSHANDI, a framework for reconstructing hand–deformable object interactions through physical simulation. To the best of our knowledge, PHYSHANDI is the first approach to achieve dense 3D reconstruction of both hands and deformable objects from sparse-view RGB-D videos.

- For deformable objet reconstruction, we propose to simulate object deformations driven by interaction forces induced from fully reconstructed 3D hand motions to achieve more accurate simulation than the existing state-of-the-art (Jiang et al., 2025) based on a sparse hand representation.

- For hand reconstruction, we refine the initial MANO (Romero et al., 2017) fitting through inverse physics, leveraging the previously reconstructed physics-based object model. To the best of our knowledge, this is the first work to show that inverse physics, guided by a physics-based deformable object model, can improve hand reconstruction.

- We achieve new state-of-the-art performance compared to our most relevant approach, PhysTwin (Jiang et al., 2025), in reconstruction, as well as future prediction and generalization to unseen interactions.

## 2 RELATED WORK

### 2.1 3D HAND-OBJECT INTERACTION MODELING

**Hand and *rigid* object interaction.** There are numerous works on modeling and reconstructing hand–object interactions from various types of inputs, e.g., RGB, depth, or RGB-D (Chen et al., 2021; Liu et al., 2021; Doosti et al., 2020; Hasson et al., 2020; 2019; Hampali et al., 2022; Tekin et al., 2019; Chen et al., 2022b; 2023), or on estimating hand-object contacts to support such reconstruction (Tse et al., 2022; Jung & Lee, 2025). Most of these methods assume a rigid object in interaction, where the

object dynamics is represented with a reference shape (e.g., a given template shape or a reconstructed shape from the first frame) with global rigid transformation $\mathbf{T} \in \mathbb{R}^{3 \times 4}$ (Chen et al., 2021; Liu et al., 2021; Doosti et al., 2020; Hasson et al., 2020; 2019; Hampali et al., 2022; Tekin et al., 2019; Chen et al., 2022b; 2023). Recently, there have been efforts to model part-wise rigid objects under hand interactions, where the object is additionally represented with part labels and per-part rigid transformations (Fan et al., 2023; Zhu et al., 2024; Zhang et al., 2025). While this enables more expressive deformation modeling than prior works with a global rigidity assumption, these methods remain non-trivial to extend to more general real-world objects (e.g., cloth, charger cables) that exhibit non-rigid deformations.

**Hand and *non-rigid* object interaction.** There are only a few works that attempt to model and reconstruct hand–*non-rigid* object interactions. HMDO (Xie et al., 2023) proposes a pipeline for markerless capture of hand–deformable object interactions from multi-view images. However, its main focus is on modeling *localized deformations driven by finger pressure*, and "*the interacting objects in [its] dataset do not have large deformations, such as 180-degree twisting or bending*" (Xie et al., 2023). Therefore, it is non-trivial to apply this method to our targeted hand–object interaction datasets, where large, global non-rigid deformations occur (e.g., bending a doll's arms). Similarly, a recent work on generating hand–deformable object interactions (Qi et al., 2025) also assumes that object deformations are locally driven by finger pressure, based on the HMDO dataset.

The most related work to ours is PhysTwin (Jiang et al., 2025), a recent state-of-the-art method for physics-based deformable object reconstruction from multi-view RGB-D videos that can handle non-localized deformations. While it addresses hand–deformable object interaction scenarios, its main focus is on object modeling, with the interacting hand represented only by sparse points sampled from input depth maps, where true hand–object contact points are unobservable due to contact occlusions. In Sec. 4, we further demonstrate that our method, with fully reconstructed hands, produces more accurate object reconstruction, and that the reconstructed object model can, in turn, effectively refine the initial hand reconstruction—mutually benefiting each other.

## 2.2 DEFORMABLE OBJECT MODELING

**Dynamic reconstruction-based modeling.** Dynamic reconstruction-based methods recover 3D representations (e.g., Occupancy Functions (Mescheder et al., 2019), Neural Radiance Fields (Mildenhall et al., 2020), 3D Gaussian Splats (Park et al., 2020)) from inputs such as RGB (Attal et al., 2023; Kratimenos et al., 2024; Li et al., 2023b; Luiten et al., 2024; Park et al., 2021a;b; Pumarola et al., 2021; Wang et al., 2023b; Xian et al., 2021; Yu et al., 2023; Tretschk et al., 2021; Chu et al., 2022), depth (Curless & Levoy, 1996; Li et al., 2008), or RGB-D (Newcombe et al., 2015) data. Most recent methods typically reconstruct a canonical representation (e.g., at the first frame) and learn deformation fields to capture object dynamics (Park et al., 2021a;b; Kratimenos et al., 2024; Xian et al., 2021). Despite differences in exact modeling approaches, they share a key limitation: the focus remains on *reconstructing* 3D representations that match observed inputs, without explicitly modeling physical properties—thereby limiting their ability to support future prediction or generalization to unseen predictions, as also discussed in (Jiang et al., 2025).

**Simulation-based modeling.** Simulation-based methods enable the modeling of object dynamics in a physically plausible manner, while also allowing generalization to unseen interactions. Early works relied on pre-scanned static objects and clean point clouds (Wang et al., 2015; Qiao et al., 2021; Du et al., 2021; Geilinger et al., 2020; Murthy et al., 2020), or were constrained to synthetic data or highly dense viewpoints (Zhang et al., 2024b; Li et al., 2023a; Chen et al., 2022a; Zhong et al., 2024; Qiao et al., 2022). More recent methods take sparse-view real RGB-D images as input (e.g., GS-Dynamics (Zhang et al., 2024a), PhysAvatar (Jiang et al., 2025)), thereby reducing the burden of expensive input capture.

## 3 PHYSHANDI: PHYSICS-BASED RECONSTRUCTION OF HAND-DEFORMABLE OBJECT INTERACTIONS

In this section, we first introduce our dense hand–deformable object interaction model based on physical simulation (Sec. 3.1). We then describe how this model can be reconstructed from sparse-view RGB-D videos (Sec. 3.2).

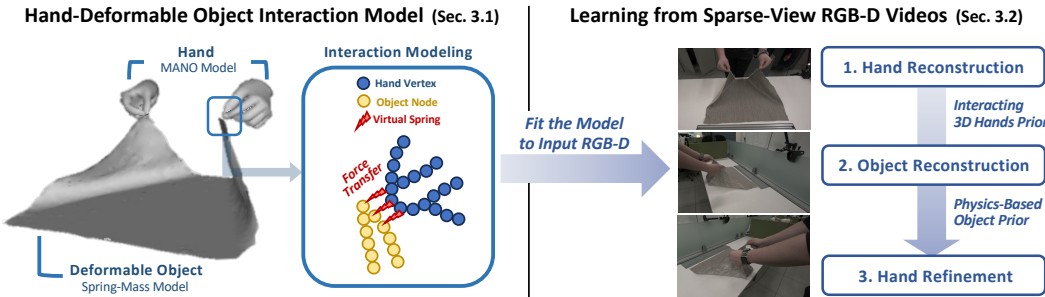

Figure 1: **PHYSHANDI models physically plausible hand–deformable object interactions.** In our interaction model, each hand is represented by the MANO model (Romero et al., 2017), and each object is represented by a spring–mass model (Liu et al., 2013). Their interaction is modeled by simulating object deformations driven by interaction forces derived from the reconstructed 3D hand motions. Our interaction model can be learned from sparse-view RGB-D videos through three stages: (1) hand reconstruction, (2) object reconstruction, and (3) hand refinement.

## 3.1 PHYSICS-BASED MODELING OF HAND AND DEFORMABLE OBJECT INTERACTIONS

We now present our approach to physically based modeling of hand–deformable object interactions. We first describe how the hand and object are *each* represented, and then elaborate on how their interaction is modeled through physical simulation.

**Hand Representation: MANO Model (Romero et al., 2017).** We represent each hand by the parameters of MANO model (Romero et al., 2017), a widely used PCA-based hand model. It maps a pose parameter $\boldsymbol{\theta} \in \mathbb{R}^{45}$, a shape parameter $\boldsymbol{\beta} \in \mathbb{R}^{10}$, a global rotation $\mathbf{R} \in SO(3)$ and a translation $\mathbf{t} \in \mathbb{R}^3$ to a dense 3D hand mesh $\mathcal{M} = (\mathcal{V}, \mathcal{F})$ with vertices $\mathcal{V} = \{\mathbf{v}_i\}_{i=1}^{778}$ and triangular faces $\mathcal{F} = \{\mathbf{f}_i\}_{i=1}^{1554}$. Since the model provides a prior that constrains the solution space of 3D hand meshes within the low-dimensional parameter space, it has been widely adopted to reduce ill-posedness in various hand reconstruction problems (e.g., interacting hand and rigid object reconstruction).

**Object Representation: Spring-Mass Model (Liu et al., 2013; Jiang et al., 2025).** We represent each deformable object using a spring-mass model (Liu et al., 2013; Jiang et al., 2025), a classical physics-based model capable of simulating the dynamic behavior of deformable objects. It models an object as a graph $\mathcal{O} = (\mathcal{N}, \mathcal{E})$. $\mathcal{N} = \{\mathbf{n}_i\}_{i=1}^{N}$ denotes a set of $N$ number of mass nodes, where each mass node $\mathbf{n}_i$ is parameterized by its position $\mathbf{x}_i \in \mathbb{R}^3$ and velocity $\mathbf{v}_i \in \mathbb{R}^{31}$, and mass $m_i \in \mathbb{R}^2$. $\mathcal{E} = \{(i, j) \mid i, j \in \{1, ..., N\}\}$ denotes a set of springs connecting the mass nodes, where $i$ and $j$ are the indices of the mass nodes connected by each spring. In this spring-mass model, each mass node can be simulated in response to the force acting on it. In particular, the force on each mass node $\mathbf{n}_i$ is modeled as:

$$\mathbf{F}_i = \sum_{(i,j) \in \mathcal{E}} \mathbf{F}_{i,j}^{\text{spring}} + \mathbf{F}_{i,j}^{\text{damping}} + \mathbf{F}_i^{\text{external}}. \qquad (1)$$

The first term $\mathbf{F}_{i,j}^{\text{spring}} = s_{ij} \left(\|\mathbf{x}_j - \mathbf{x}_i\| - r_{ij}\right) \frac{\mathbf{x}_j - \mathbf{x}_i}{\|\mathbf{x}_j - \mathbf{x}_i\|}$ represents the spring force between the connected mass nodes $\mathbf{n}_i$ and $\mathbf{n}_j$ based on Hooke's law, where $s_{ij}$ is the stiffness parameter, and $r_{ij}$ is the rest length of the spring $(i, j)$. This term encourages the spring-mass system to maintain the rest length of each spring. The second term $\mathbf{F}_{i,j}^{\text{damping}} = -\gamma_{ij}(\mathbf{v}_i - \mathbf{v}_j)$ is a dashpot damping force between $\mathbf{n}_i$ and $\mathbf{n}_j$, where $\gamma_{ij}$ is the dashpot damping coefficient of the spring $(i, j)$. It penalizes relative velocity along the spring direction, stabilizing the system and preventing oscillations. The final term $\mathbf{F}_i^{\text{external}}$ models external forces acting on the mass node, such as gravity.

Given the force $\mathbf{F}_i$ computed from the above modeling equation (Eq. 1) at each time $t$, the updated position $\mathbf{x}_i$ of node $i$ at time $t + 1$ is obtained by numerically integrating Newton's second law over time, such that $\mathbf{v}_i^{t+1} = \mathbf{v}_i^t + \Delta t \frac{\mathbf{F}_i}{m_i}$ and $\mathbf{x}_i^{t+1} = \mathbf{x}_i^t + \Delta t\, \mathbf{v}_i^{t+1}$.

---

[1]While the hand mesh vertex is also denoted by $\mathbf{v}_i$, we allow a slight abuse of notation to remain consistent with notation conventions used in related work.

[2]Directly following prior work (Jiang et al., 2025), we assign a unit mass to all nodes in the spring–mass system, since ground-truth mass values are not available in our setting.

**Hand-Deformable Object Interaction Modeling.** Given these hand and deformable object representations, we now describe how their interaction is modeled by simulating object deformation with a spring–mass system, driven by forces induced by the motions of MANO hand meshes. We follow the common strategy for modeling interaction forces in the spring–mass system (Liu et al., 2013), where *virtual springs* are connected between object nodes and the interactee (MANO hand vertices in our case) that are detected to be in contact within a connection radius $\delta$ (left subfigure of Fig. 1).

Formally, in our final spring–mass system, the mass nodes are defined as $\mathcal{N} \cup \mathcal{V}'$, which is the superset of the object nodes $\mathcal{N}$ and the *virtual hand nodes* $\mathcal{V}'$. Since these virtual hand nodes are used to induce forces for simulating object deformation, their positions and velocities are determined by the tracked MANO vertices $\mathcal{V}$, and then fixed as a boundary condition throughout the simulation. The virtual springs are then defined as $\mathcal{E} \cup \mathcal{E}^{\text{virtual}}$, which is the superset of the object springs $\mathcal{E}$ and the virtual springs $\mathcal{E}^{\text{virtual}}$ connecting the contacted object and hand nodes. Note that, during simulation, these virtual springs encourage the object regions in contact with the hand to smoothly deform according to the fixed hand vertex motion, as the spring and dashpot damping forces in the spring–mass system ($\mathbf{F}_{i,j}^{\text{spring}}$ and $\mathbf{F}_{i,j}^{\text{damping}}$ in Eq. 1) act to maintain the contact topology (i.e., the rest length of the *virtual springs* between hand vertices and object nodes). This ensures object dynamics that are physically plausible and coherent with the interacting hand movements, while being capable of modeling large and complex non-rigid deformations.

## 3.2 Learning Our Interaction Model from Sparse-View RGB-D Videos

We now explain how the physics-based hand–deformable object interaction model described in Sec. 3.1 can be reconstructed from sparse-view RGB-D video inputs. Our learning pipeline consists of three stages: (1) hand reconstruction, (2) object reconstruction, and (3) hand refinement.

**Hand Reconstruction.** In this stage, we fit the MANO hand model (Romero et al., 2017) to multi-view RGB-D videos. Our optimization target for each frame is the MANO parameters $\Theta_{hand} = \{\boldsymbol{\theta}, \boldsymbol{\beta}, \mathbf{R}, \mathbf{t}\}$, where $\boldsymbol{\theta} \in \mathbb{R}^{45}$ and $\boldsymbol{\beta} \in \mathbb{R}^{10}$ are pose and shape parameters, and $\mathbf{R} \in SO(3)$ and $\mathbf{t} \in \mathbb{R}^3$ are global rotation and translation. Our optimization objective is formulated as:

$$\min_{\Theta_{hand}} \mathcal{L}_{2D}(\Theta_{hand}, \mathbf{U}) + \lambda_{\text{depth}} \mathcal{L}_{\text{depth}}(\Theta_{hand}, \mathbf{D}) + \lambda_{\text{temp}} \mathcal{L}_{\text{temp}}(\Theta_{hand}, \Theta_{hand}^{prev}), \tag{2}$$

where $\mathcal{L}_{2D}$ measures the reprojection error between the projected MANO keypoints and the 2D keypoint supervision $\mathbf{U} \in \mathbb{R}^{V \times 21 \times 2}$ for each of the $V$ views. $\mathcal{L}_{\text{depth}}$ measures the discrepancy between the rendered MANO depth and the observed depth maps $\mathbf{D} \in \mathbb{R}^{V \times H \times W}$, where $H$ and $W$ denote the depth map resolution, and $\mathcal{L}_{\text{temp}}$ regularizes the temporal smoothness of the MANO parameters with respect to those fitted in the previous frame, $\Theta_{hand}^{prev}$. The coefficients $\lambda_{2D}$, $\lambda_{\text{depth}}$, and $\lambda_{\text{temp}}$ control the relative weight of each loss term.

**Object Reconstruction.** Conditioned on the fitted 3D hands, we now fit the spring–mass model representing the deformable object under interaction. For this stage, we mainly follow the object model fitting pipeline of PhysTwin (Jiang et al., 2025), though it does not consider the dense hand vertices as the interactee. At a high level, the object's 3D geometry at $t = 0$ is first obtained using an image-to-3D generative model (Xiang et al., 2025). The object dynamics for $t \in [1, T]$ are then simulated with a spring–mass system, and the physical parameters (e.g., $s_{ij}$, $\gamma_{ij}$ and $\delta$ in Sec. 3.1) are optimized so that the simulated geometries better match the input observations. The optimization objective is formulated as minimizing two terms: (1) $\mathcal{L}_{chamfer}$, which measures the Chamfer distance between the simulated node positions and the observed 3D point clouds lifted from the input depth maps, and (2) $\mathcal{L}_{track}$, an $\ell_2$ loss between the simulated node positions and the pseudo–ground-truth 3D points tracked by CoTracker3 (Karaev et al., 2024).

While we kindly refer the reader to our supplementary material or Jiang et al. (2025) for more details on this stage, we highlight a key difference in our spring–mass simulations: our approach models interaction forces from the dense 3D hand geometry fitted with the MANO model in the previous stage, whereas PhysTwin approximates these forces using sparse points sampled from input depth maps, where true hand–object contact points are unobservable due to contact occlusions. As this may limit the precision of interaction force modeling during simulation, our method achieves more accurate simulation results, as discussed in Sec. 4. In the supplementary, we also provide numerical

analysis showing that our dense hand interactee enables the spring–mass model topology to be reconstructed more optimally than PhysTwin in the view of peridynamics (Silling & Askari, 2005; Silling et al., 2007; Wang et al., 2023a).

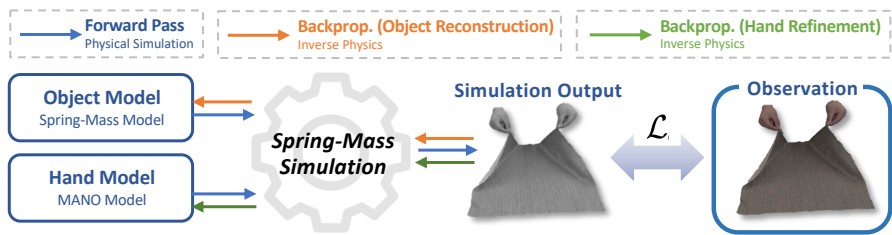

Figure 2: **Illustration of inverse physics for object reconstruction and hand refinement.** Our spring–mass simulation is driven by the spring–mass object model and the MANO hand model. In the object reconstruction stage, the object model is fitted via inverse physics given the initial MANO models, while in the subsequent hand refinement stage, the initial MANO models are refined given the reconstructed object model. $\mathcal{L}$ denotes our loss function, composed of $\mathcal{L}_{chamfer}$ and $\mathcal{L}_{track}$.

**Hand Refinement.** After the object reconstruction stage, we can leverage the reconstructed physics-based object model as an additional prior to further refine the initial hand model fitting, enforcing it to produce *object simulations* better aligned with the input observation. Specifically, we reuse the same $\mathcal{L}_{chamfer}$ and $\mathcal{L}_{track}$ losses from the object reconstruction stage to measure the discrepancy between the simulated object nodes, and the ground-truth observations at each timestep $t$. In this stage, however, we apply them to fine-tune the MANO model parameters via inverse physics (see Fig. 2), using gradient-descent-based optimization. Let $\mathcal{S}_t(\cdot)$ denote the function that returns the simulated object nodes at timestep $t$ given the MANO hand parameters. The refined hand parameters $\tilde{\Theta}_{hand}$ are then optimized as:

$$\tilde{\Theta}_{hand} = \arg\min_{\Theta_{hand}} \frac{1}{T} \sum_{t=1}^{T} \mathcal{L}_{\text{chamfer}}(\mathcal{S}_t(\Theta_{hand}), \mathcal{P}) + \lambda_{track}\mathcal{L}_{\text{track}}(\mathcal{S}_t(\Theta_{hand}), \mathbf{T}), \quad (3)$$

where $\mathcal{P}$ and $\mathbf{T}$ denote the ground-truth lifted point cloud and tracked points, respectively. This inverse-physics–based refinement is particularly effective when the hand observation is highly ill-posed; while we empirically find that the initial hand reconstruction stage is already sufficient to achieve state-of-the-art results with multi-view RGB-D inputs, this hand refinement stage proves especially effective when inference is performed from sparser inputs (e.g., future prediction in a single-view setting). To the best of our knowledge, this is the first work to demonstrate that hand model fitting accuracy can be improved through the inverse physics of deformable object simulation.

## 4 EXPERIMENTS

In this section, we experimentally evaluate the effectiveness of our method. We first describe our experimental settings in Sec. 4.1, and then present the comparison results in Sec. 4.2.

### 4.1 EXPERIMENT SETTINGS

**Dataset.** We mainly use the PhysTwin dataset (Jiang et al., 2025), which provides 22 three-view RGB-D videos of hand–deformable object interactions. Furthermore, since most sequences in PhysTwin involve only sparse hand–object contacts (e.g., fingers pinching the object), we also present supplementary results on 19 newly collected sequences with denser hand–object contacts as shown in Fig. 3, which we refer to as the DENSEHDI dataset. We collected this dataset using the same data collection protocol as Jiang et al. (2025), including 10 additional objects (e.g., pouch, towel, paper cup, hat) (please refer to the supplementary for more details). Upon publication, we will release this dataset to facilitate future research on modeling hand–deformable object interactions.

**Baseline and Tasks.** We primarily compare against PhysTwin (Jiang et al., 2025), the current state-of-the-art in physics-based object reconstruction from sparse-view RGB-D videos. In addition, we include two further baselines, Spring-Gaus (Zhong et al., 2024) and GS-Dynamics (Zhang et al.,

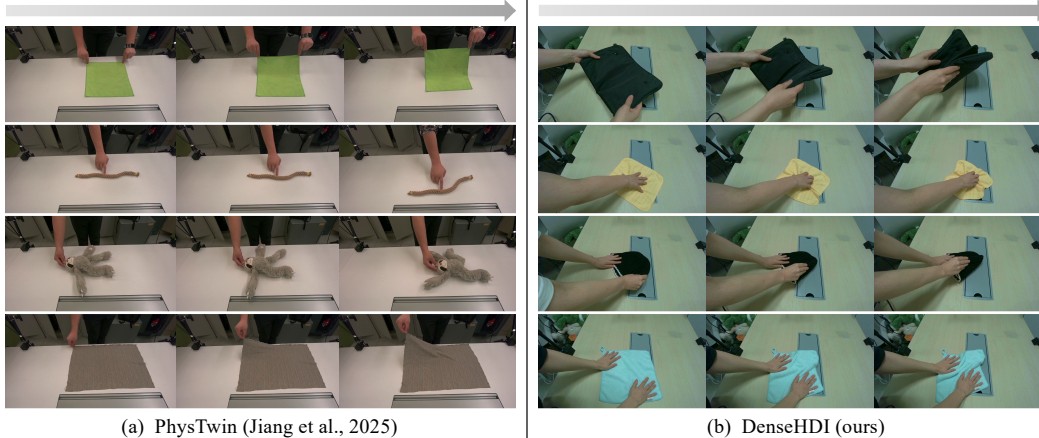

(a) PhysTwin (Jiang et al., 2025)        (b) DenseHDI (ours)

Figure 3: **Captured sequences in PhysTwin (Jiang et al., 2025) and DENSEHDI.** The sequences in DENSEHDI feature denser hand–object contacts.

2024a), following the comparative protocol of PhysTwin. For Spring-Gaus, we use the controller-augmented variant used in PhysTwin, as its original formulation does not support external control inputs. We follow the evaluation convention established by this prior work and evaluate on three tasks: (1) reconstruction & resimulation, (2) future prediction, and (3) generalization to unseen interactions, each of which will be discussed in detail in the following sections.

**Evaluation metrics.** We directly adopt the evaluation metrics used in PhysTwin (Jiang et al., 2025) to ensure fair comparisons. In particular, we use metrics that evaluate geometric and photometric discrepancies between the reconstructed objects and the ground truth—either in 3D space (Chamfer Distance, Tracking Error) or in projected 2D space (IoU, PSNR, SSIM, and LPIPS). To enable the use of photometric metrics (PSNR, SSIM, and LPIPS), we follow PhysTwin by learning surrogate Gaussian splats bounded to the object model, which allows rendering of the reconstructed object dynamics for evaluation. We also note that the optimization-based reconstruction pipelines in both PhysTwin (Jiang et al., 2025) and our method (Sec. 3.2) involve stochasticity (e.g., random parameter initialization). Therefore, we run the official implementation of PhysTwin and compare the average results over 10 runs for more reliable comparisons.

In addition, we observed that several sequences in PhysTwin dataset contain large static regions while only a subset of points undergo meaningful deformation (e.g., the cloth region fixed on the table in Fig. 4(a). To more clearly measure accuracy in the dynamically deforming regions—which are the primary focus of physics-based reconstruction—we report two versions of the Chamfer Distance for the PhysTwin benchmark experiments. $CD_{full}$ is computed over all object points as in PhysTwin (Jiang et al., 2025), whereas $CD_{dyn}$ evaluates only the points that exhibit non-negligible motion. Formally, given pseudo-ground-truth point trajectories $x_{1:T}$ obtained from CoTracker and depth input, we include a point in the dynamic set if $\|x_i - x_1\|^2 > \tau_{dyn}$, where $\tau_{dyn}$ is a motion-magnitude threshold. $CD_{dyn}$ therefore focuses the evaluation on regions where true deformation occurs, providing a more informative measure of dynamic reconstruction quality.

## 4.2 EXPERIMENTAL COMPARISONS

### 4.2.1 RECONSTRUCTION & RESIMULATION

In the reconstruction and resimulation experiments, we evaluate reconstruction accuracy on the *seen* frames used during physics-based object model fitting, following the protocol in Jiang et al. (2025). Tab. 1 (left) presents our quantitative results on the PhysTwin dataset, where the upper subtable reports the *results on the dense interaction sequences* (excluding those with only sparse hand–object contacts such as pinching), and the lower subtable shows the *results on the full sequences*. Our method outperforms Spring-Gaus and GS-Dynamics by a large margin across all metrics, and also outperforms PhysTwin on most metrics. It is particularly effective on dense interaction sequences, demonstrating its strength in modeling hand–object interactions involving more complex contacts. Although our method achieves comparable results with PhysTwin on a few metrics, the qualitative

comparisons in Fig. 4 and our supplementary video clearly show that it produces more accurate reconstruction and resimulation. Moreover, the newly introduced $CD_{dyn}$ metric reveals a much clearer performance gap between the two methods: because $CD_{full}$ averages positional discrepancies over both deforming and static regions, it cannot faithfully capture reconstruction accuracy in the deformable parts. The larger margins observed in $CD_{dyn}$ therefore highlight our method's advantages in modeling dynamically deforming object regions. Indeed, during simulation, PhysTwin approximates interaction forces from sparse points (whose cardinality is around 30) sampled from depth maps without full 3D hand modeling. This may lead to less accurate simulations, as it (1) limits the precision of interaction force modeling—since actual hand–object contact points cannot be fully observed from depth sensors due to mutual occlusion—and (2) results in suboptimal model topology reconstruction for simulation, as analyzed in the supplementary. Beyond PhysTwin, we also observe that Spring-Gaus (Zhong et al., 2024) and GS-Dynamics (Zhang et al., 2024a) perform significantly worse than both our method and PhysTwin, consistent with the findings reported in Jiang et al. (2025). Spring-Gaus was originally proposed for settings with considerably denser viewpoints, and under our three-view sparse-input configuration its simulation becomes unstable, leading to noticeably broken or collapsed object geometry, as illustrated in Fig. 4. Similarly, GS-Dynamics is designed to leverage long motion sequences through its GNN-based motion representation. Under our benchmark configuration—where each object provides only 30–200 training frames—the model fails to capture meaningful deformation behavior and instead learns only subtle motions, as reflected in the qualitative results. These observations further underline the challenges that existing physics-based or GNN-based motion methods face in sparse-view, short-sequence hand–object interaction scenarios.

In Tab. 2 (left), we additionally report results on our DENSEHDI dataset, which mainly captures dense hand–object interactions. Here, our method outperforms the baseline on all metrics, further validating its effectiveness in modeling dense hand–object interactions through full 3D hand modeling.

| Method | Reconstruction & Resimulation | | | | | | | Future Prediction | | | | | | |
| | 3D Metrics | | | 2D Metrics | | | | 3D Metrics | | | 2D Metrics | | | |
| | $CD_{full}\downarrow$ | $CD_{dyn}\downarrow$ | Track Err. $\downarrow$ | IoU $\uparrow$ | PSNR $\uparrow$ | SSIM $\uparrow$ | LPIPS $\downarrow$ | $CD_{full}\downarrow$ | $CD_{dyn}\downarrow$ | Track Err. $\downarrow$ | IoU $\uparrow$ | PSNR $\uparrow$ | SSIM $\uparrow$ | LPIPS $\downarrow$ |
| (a) Results on the dense interaction sequences. | | | | | | | | | | | | | | |
| Spring-Gaus Zhong et al. (2024) | 38.84 | 27.79 | 4.65 | 0.55 | 21.41 | 0.94 | 7.80 | 56.51 | 37.38 | 7.69 | 0.42 | 19.91 | **0.94** | 9.27 |
| GS-Dynamics Zhang et al. (2024a) | 13.67 | 33.37 | 1.81 | 0.74 | 23.12 | 0.94 | 3.80 | 33.99 | 56.79 | 4.50 | 0.51 | 18.97 | 0.93 | 7.79 |
| PhysTwin Jiang et al. (2025) | 5.90 | 10.78 | 1.00 | 0.84 | 25.23 | **0.95** | 2.65 | 11.45 | 16.32 | 2.10 | 0.70 | 22.07 | **0.94** | 5.38 |
| PHYSHANDI (Ours) | **5.30** | **8.32** | **0.89** | **0.85** | **25.62** | **0.95** | **2.52** | **10.57** | **14.35** | **2.05** | **0.73** | **22.84** | **0.94** | **4.83** |
| (b) Results on the full sequences. | | | | | | | | | | | | | | |
| Spring-Gaus Zhong et al. (2024) | 33.60 | 26.39 | 4.07 | 0.62 | 21.24 | 0.94 | 8.09 | 46.54 | 49.29 | 6.61 | 0.48 | 19.59 | 0.93 | 9.44 |
| GS-Dynamics Zhang et al. (2024a) | 13.79 | 24.73 | 2.18 | 0.72 | 24.01 | **0.95** | 4.14 | 38.84 | 52.96 | 6.88 | 0.46 | 19.38 | 0.94 | 8.32 |
| PhysTwin Jiang et al. (2025) | 5.52 | 7.63 | 0.97 | **0.84** | 26.32 | **0.95** | 2.88 | 12.26 | 14.42 | 2.44 | **0.69** | 22.80 | **0.95** | 5.15 |
| PHYSHANDI (Ours) | **5.40** | **7.30** | **0.96** | **0.84** | **26.44** | **0.95** | **2.87** | **12.04** | **13.63** | **2.41** | 0.68 | **22.96** | **0.95** | **5.02** |

Table 1: **Reconstruction & Resimulation and Future Prediction results on the PhysTwin dataset (Jiang et al., 2025)**. Our method outperforms the state-of-the-art (Jiang et al., 2025) on most metrics, and is particularly effective in modeling hand–object interactions involving dense contacts. CD is measured in millimeters, and Track Err. and LPIPS are scaled by $\times 100$ for readability.

### 4.2.2 FUTURE PREDICTION

In the future prediction experiments, we evaluate reconstruction quality on future frames that were *unseen* during physics-based object model fitting.

**Three-View RGB-D Inputs.** In Tables 1 and 2 (right), we present future prediction results on the PhysTwin and DENSEHDI datasets, respectively. Our method surpasses Spring-Gaus (Zhong et al., 2024) and GS-Dynamics (Zhang et al., 2024a) by a substantial margin across all metrics and further exceeds the state-of-the-art (Jiang et al., 2025) on most metrics, demonstrating its effectiveness on future prediction as well. Our qualitative comparisons in Fig. 4 and our supplementary video also show that our approach produces future predictions that are more accurately aligned with the ground-truth observations.

**Single-View RGB-D Inputs.** We additionally report future prediction results on *single-view RGB-D inputs*, which represent a more challenging scenario than the multi-view setting considered in the existing state-of-the-art (PhysTwin (Jiang et al., 2025)). In this setting, PhysTwin must approximate interaction forces from sparse hand points sampled from a *single-view* depth map, which are highly partial. We empirically observed that its object simulation frequently fails due to errors in identifying hand–object contact points (determined by the threshold $\delta$ in Sec. 3.1), and therefore could not be included in our comparisons. As shown in Tab. 3, our method robustly addresses the challenging task

| Method | Reconstruction & Resimulation | | | | | | Future Prediction | | | | | |
|---|---|---|---|---|---|---|---|---|---|---|---|---|
| | 3D Metrics | | 2D Metrics | | | | 3D Metrics | | 2D Metrics | | | |
| | CD ↓ | Track Err. ↓ | IoU ↑ | PSNR ↑ | SSIM ↑ | LPIPS ↓ | CD ↓ | Track Err. ↓ | IoU ↑ | PSNR ↑ | SSIM ↑ | LPIPS ↓ |
| PhysTwin Jiang et al. (2025) | 5.59 | 1.58 | 0.76 | 21.22 | 0.94 | 7.34 | 7.98 | 2.42 | 0.63 | **19.76** | **0.94** | 8.56 |
| **PHYSHANDI (Ours)** | **5.06** | **1.50** | **0.78** | **21.61** | **0.95** | **7.08** | **7.54** | **2.40** | **0.65** | 19.75 | 0.94 | **8.46** |

Table 2: **Reconstruction & Resimulation and Future Prediction results on the DENSEHDI dataset**. Our method outperforms the state-of-the-art (Jiang et al., 2025) on most metrics, demonstrating its effectiveness. CD is measured in millimeters, and Track Err. and LPIPS are scaled by ×100 for readability.

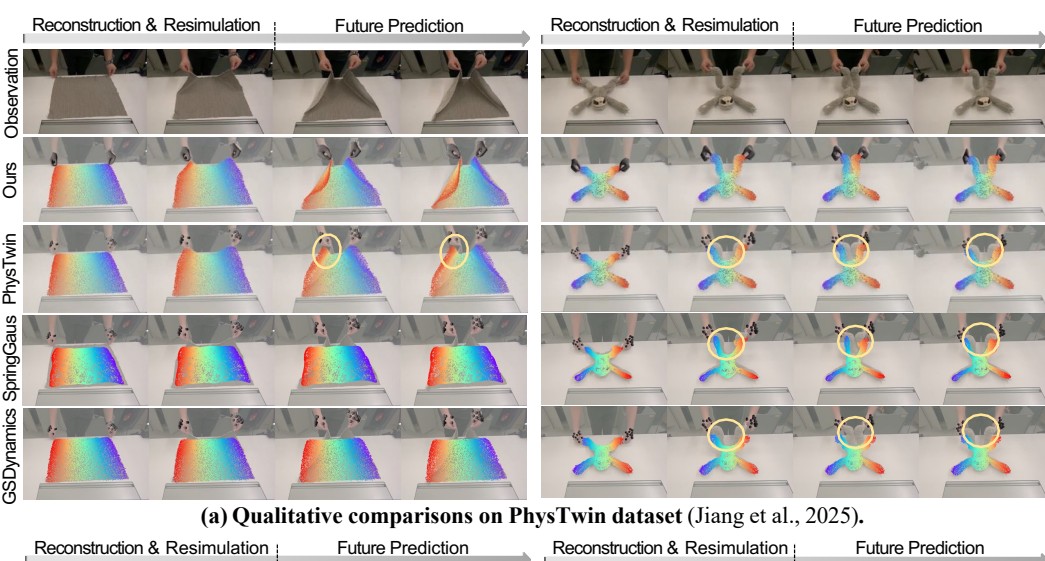

(a) **Qualitative comparisons on PhysTwin dataset** (Jiang et al., 2025)**.**

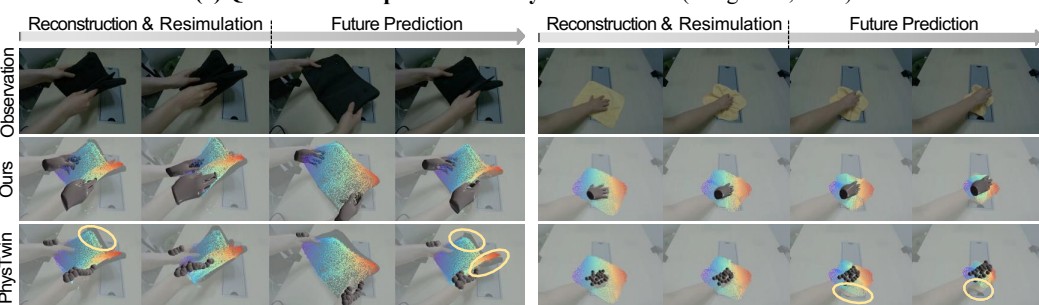

(b) **Qualitative comparisons on DenseHDI dataset.**

Figure 4: **Qualitative comparisons on (1) reconstruction and resimulation, and (2) future prediction.** Yellow circles indicate regions where object simulations are less accurately aligned with the ground-truth observations or with the interacting hand contacts. Compared to all the baselines, our method produces more accurate object simulations.

of single-view future prediction by leveraging full 3D reconstruction from partial inputs. The table also compares our results without inverse physics-based hand refinement, where our refinement is shown to be significantly effective when the input view is highly sparse to sufficiently constrain hand model fitting.

To further evaluate hand fitting accuracy, we additionally report the metric Hand CD in the table, which measures the Chamfer Distance between the fitted hand meshes and the ground-truth 3D hand point cloud lifted from the multi-view depth maps available in the dataset.[3] Our hand refinement also noticeably improves this hand fitting accuracy metric; to the best of our knowledge, this is the first work to demonstrate that a physics-based deformable object prior can benefit hand reconstruction.

---

[3]Note that in other multi-view experiments, these multi-view depth maps are used as *inputs* during training and are therefore not treated as ground truth for evaluation.

| Method | 3D Metrics | | | 2D Metrics | | | |
|---|---|---|---|---|---|---|---|
| | CD ↓ | Track Err. (×0.01) ↓ | Hand CD ↓ | IoU ↑ | PSNR ↑ | SSIM ↑ | LPIPS (×0.01) ↓ |
| **PHYSHANDI (Ours) - Hand Ref.** | 42.8 | 7.36 | 7.57 | 0.49 | 19.50 | **0.94** | 8.25 |
| **PHYSHANDI (Ours)** | **33.5** | **6.75** | **7.17** | **0.51** | **19.67** | 0.94 | **8.13** |

Table 3: **Single-view future prediction results on the PhysTwin dataset (Jiang et al., 2025).** Our hand refinement using the physics-based object prior is effective in enhancing both object and hand reconstruction quality.

### 4.2.3 GENERALIZATION TO UNSEEN INTERACTIONS

In the generalization to unseen interactions experiments, we evaluate reconstruction quality on novel interaction sequences performed on the same object but with different interaction types, following the evaluation protocol of (Jiang et al., 2025). Our method achieves superior results on all metrics (except SSIM, where it matches the baseline), demonstrating its strong generalizability to unseen interaction actions.

| Method | 3D Metrics | | 2D Metrics | | | |
|---|---|---|---|---|---|---|
| | CD ↓ | Track Err. (×0.01) ↓ | IoU ↑ | PSNR ↑ | SSIM ↑ | LPIPS (×0.01) ↓ |
| PhysTwin (Jiang et al., 2025) | 8.94 | 1.77 | 0.79 | 25.44 | **0.94** | 3.42 |
| **PHYSHANDI (Ours)** | **8.38** | **1.70** | **0.82** | **25.89** | 0.94 | **3.40** |

Table 4: **Generalization to unseen interactions results on the PhysTwin dataset (Jiang et al., 2025).** Our method demonstrates superior generalizability compared to the state of the art (Jiang et al., 2025).

## 5 DISCUSSIONS & LIMITATIONS

**Evaluation of multi-view hand reconstruction.** Although we directly evaluated hand reconstruction accuracy in the single-view setting (Sec. 4.2.2) using ground-truth hand point clouds lifted from multi-view depth maps, this evaluation is not possible in the main multi-view experiments since those depth maps are used as inputs during training and more precise hand annotations are unavailable. Indeed, in prior work (e.g., (Hampali et al., 2020)), MANO fitting to multi-view RGB-D is often treated as a way to *annotate ground-truth 3D hand meshes* in datasets lacking such labels, and fitting quality is typically assessed indirectly via downstream applications. Motivated by this, we evaluate hand fitting quality primarily through physics-based deformable object reconstruction, where more accurate hands directly yield more accurate object simulations. Nonetheless, a direct evaluation would be valuable—for example, by capturing *denser-view ground-truth* 3D hands and comparing them against our sparse multi-view reconstructions, if such a capture system is available.

**Handling dynamic contact changes.** As discussed in Sec. 3.1, our interaction force is modeled to encourage the maintenance of the contact topology (i.e., the rest length of the virtual springs between hand vertices and object nodes). This modeling assumes that the hand–object contact topology remains static within a sequence. While this assumption held for the benchmarks we considered (PhysTwin and DENSEHDI), it may not hold for sequences with dynamic hand–object contact changes, making the handling of such dynamics an important direction for future research. In addition, such interaction force modeling does not account for the actual force (e.g., finger pressure) but instead serves as a boundary condition to drive the simulation of the spring–mass model. Explicitly modeling the actual hand force would be non-trivial, yet an interesting future research direction with potential applications in haptics.

## 6 CONCLUSION

We presented PHYSHANDI, a physics-based framework for modeling and reconstructing hand–object interactions involving highly non-rigid objects. By incorporating physics-based priors and simulating object deformations driven by forces from the fully reconstructed 3D hands, our method produces reconstructions that are both physically plausible and consistent with interacting hand dynamics. Through a reconstruction pipeline based on sparse-view RGB-D inputs, PHYSHANDI demonstrates superior performance over existing baselines in reconstruction, future prediction, and generalization to unseen interactions. We believe this work takes a step toward more general and robust modeling of everyday hand–object interactions, opening up new opportunities for applications in embodied AI and digital human modeling.

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

## A    COMPARATIVE ANALYSIS OF SPRING–MASS MODEL TOPOLOGY

In this section, we further discuss why our object simulation may achieve better performance than the current state-of-the-art, PhysTwin (Jiang et al., 2025), as empirically shown in Sec. 4. Specifically, we examine differences in the optimized spring-mass model topology, since *"the behavior of the [spring-mass] model is dependent on the topology"* (Nealen et al., 2006). As discussed in Sec. 3, the connection radius $\delta$ is the primary factor defining the model topology in both our method and PhysTwin, thus we present a numerical analysis of the fitted $\delta$.

In particular, we refer to practical analyses of particle-based models with radius-limited neighbor interactions (e.g., peridynamics) (Silling & Askari, 2005; Silling et al., 2007; Wang et al., 2023a), as our spring-mass model likewise restricts interactions to neighbors within a distance cutoff $\delta$. These studies show that, given the spatial discretization resolution $\Delta x$ of the object[4], the ratio $\delta/\Delta x$ should remain close to a small constant $r$, and that *"values much larger than this may result in excessive wave dispersion and require very large computer run times."* (Silling & Askari, 2005)

Inspired by this, we introduce a simple measure of deviation from this recommended ratio of connection radius to discretization resolution. Specifically, we report the **Radius-to-Resolution Deviation ($RRD$)**,

$$RRD = |(\delta/\Delta x)/r - 1|, \tag{4}$$

where we use $r = 3$ as the reference factor, reflecting a value commonly acknowledged as plausible in the cited guidelines. As shown in Tab. 5, our method yields about $2\times$ lower $RRD$ for object springs and over $7\times$ lower $RRD$ for virtual springs compared to PhysTwin (Jiang et al., 2025), indicating that our model topology is more optimal according to the analyses in the aforementioned literature.

We also note that these results relate to the topology and contact visualization in Fig. 5, where PhysTwin's sparser control points are likely to result in excessively long virtual-spring lengths to maintain coverage, whereas our denser hand reconstruction precisely localizes contacts without unnecessarily elongating the springs. In addition, the control force visualization (right column of Fig. 5) shows that PhysTwin's wider virtual-spring coverage diffuses forces over a larger area, weakening local actuation around the true contact. In contrast, ours concentrates forces only where contact actually occurs, which is preferable for fine, localized manipulation. Together, these analyses suggest that dense hand reconstruction enhances the topology optimization of the spring–mass model, yielding a smaller, resolution-matched connection radius and more reliable dynamics.

| Method | $RRD_{object} \downarrow$ | $RRD_{virtual} \downarrow$ |
|---|---|---|
| PhysTwin (Jiang et al., 2025) | 0.64 | 2.63 |
| **PHYSHANDI** (Ours) | **0.32** | **0.35** |

Table 5: **Radius-to-Resolution Deviation ($RRD$)** for object and virtual springs (lower is better). $RRD = |(\delta/\Delta x)/r - 1|$. Our method achieves about $2\times$ lower $RRD$ for object springs and over $7\times$ lower for virtual springs compared to PhysTwin (Jiang et al., 2025).

## B    DATASET DETAILS

In this section, we present the details of our newly captured dataset, DENSEHDI, introduced in Sec. 4.1. For data acquisition and pre-processing, we follow the same protocol as PhysTwin (Jiang et al., 2025), using three RealSense D455 RGB-D cameras to record three-view videos of hand–deformable object interactions. In total, we collect 19 sequences, each lasting 2–8 seconds, spanning 10 object types (e.g., swimming cap, cloth, pouch, towel). The dataset includes diverse interactions, such as folding a pouch or towel and squeezing a cloth. We note that the existing PhysTwin dataset (Jiang et al., 2025) primarily captures sparse, point-like hand–object contacts (e.g., pointing at or pushing with one finger, or pinching with two fingers), whereas our dataset focuses on capturing denser hand–object contacts, such as wiping with a dishcloth or folding a pouch using the palm. Visualizations of these captured sequences are provided in Fig. 3.

---

[4]In our case, $\Delta x$ is approximated as the mean distance to each node's four nearest neighbors, averaged across all nodes.

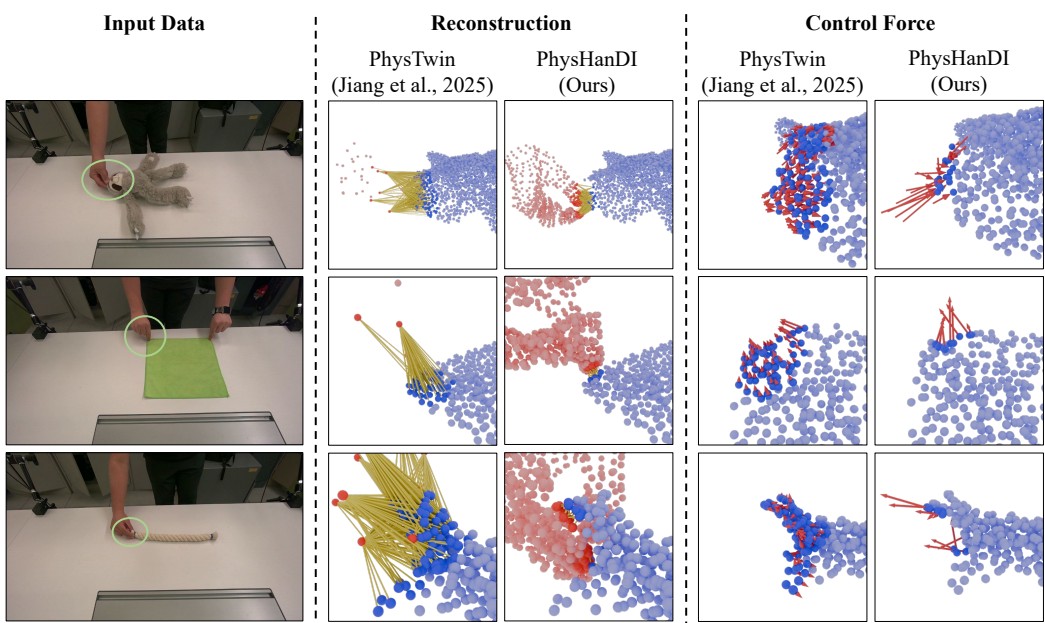

| | **Input Data** | **Reconstruction** | | **Control Force** | |
| | | PhysTwin (Jiang et al., 2025) | PhysHanDI (Ours) | PhysTwin (Jiang et al., 2025) | PhysHanDI (Ours) |

Figure 5: **Comparisons in reconstructed spring–mass model topology and force.** (1) *Topology reconstruction.* PhysTwin (Jiang et al., 2025)'s sparser hand points tend to result in excessively long virtual spring lengths to maintain contact coverage, whereas ours based on dense hand points precisely localizes contacts without unnecessary spring elongation—considered a more optimal topology in prior works (Silling & Askari, 2005; Silling et al., 2007; Wang et al., 2023a). (2) *Control force visualization.* PhysTwin's broader spring coverage disperses forces across non-contact regions, whereas ours concentrates forces at the actual contact, favoring local, detailed manipulation. *Visualization key.* Yellow line segments depict virtual-springs $\mathcal{E}^{\text{virtual}}$. Blue spheres denote object nodes $\mathcal{N}$ and red spheres denote control nodes $\mathcal{V}'$. Red arrows visualize control-force vectors induced on object nodes by virtual springs.

## C METHOD DETAILS

In this section, we provide additional details on reconstructing our dense hand–deformable object interaction model from sparse-view RGB-D video inputs, as discussed in Sec. 3.2.

### C.1 HAND RECONSTRUCTION

In the *hand reconstruction* stage, we fit the MANO model (Romero et al., 2017) to the input multi-view RGB-D videos using the loss function defined in Eq. 2 (Sec. 3.2). $\mathcal{L}_{2D}$, $\mathcal{L}_{depth}$, and $\mathcal{L}_{temp}$ are defined as L2 losses, with $\lambda_{depth}$ and $\lambda_{temp}$ set to $1 \times 10^2$ and $5 \times 10^5$, respectively.

To obtain 2D keypoint supervision for computing $\mathcal{L}_{2D}$, we use an off-the-shelf estimator (MediaPipe (Lugaresi et al., 2019)). However, we empirically observe that it yields missing or implausible predictions for heavily occluded hand joints, which degrade the MANO fitting results—particularly in our sparse-view setting. To mitigate this, we additionally obtain 2D keypoint supervision $\mathbf{U}^{\text{mano}}$ from a *monocular MANO parameter estimator* (Dong et al., 2024), which yields plausible predictions constrained by the MANO space, though with less precise 2D alignment.[5] We empirically find that combining discrepancy losses with respect to $\mathbf{U}^{2D}$ (from MediaPipe (Lugaresi et al., 2019)) and $\mathbf{U}^{\text{mano}}$ yields more robust MANO fitting, with the loss weight for $\mathbf{U}^{\text{mano}}$ set to 0.5.

---

[5] See related discussions in prior works, e.g., Li et al. (2021). Although the MANO-based estimator predicts full 3D hand shapes and poses, we use only its 2D projections since its depth estimates are ambiguous due to the *monocular* setting (e.g., projective ambiguity, scale–depth trade-off).

For optimizing the MANO parameters based on the aforementioned loss, we use the AdamW optimizer (Loshchilov & Hutter, 2017) for 1500 steps with a learning rate of $2 \times 10^{-3}$, decaying by a factor of 0.98 every 40 steps. The MANO parameters at each frame are initialized from the fitting results of the previous frame, while the first frame is initialized randomly.

## C.2 Object Reconstruction

After the hand reconstruction stage, we fit the spring–mass model (Liu et al., 2013; Jiang et al., 2025) representing the deformable object, conditioned on the previously fitted 3D hands. Directly following Jiang et al. (2025), we adopt a hierarchical optimization scheme with (1) a **sparse (zero-order)** stage followed by (2) a **dense (first-order)** stage.

**Sparse (zero-order) stage.** We optimize the coarse, non-differentiable spring–mass model parameters $\Theta_0 = \{\mathcal{T}, s_{\text{global}}, \eta\}$, where $\mathcal{T}$ denotes the spring–mass topology parameterized by a connection radius $\delta$ and a maximum number of connected nodes $d_{\max}$, $s_{\text{global}}$ is the global spring stiffness (assuming homogeneity at this stage), and $\eta$ represents collision parameters.

**Dense (first-order) stage.** With $\Theta_0$ fixed, we refine the differentiable per-spring parameters $\Theta_1 = \{s_{ij}, \gamma_{ij}\}_{(i,j) \in \mathcal{E} \cup \mathcal{E}^{\text{virtual}}}$, where $s_{ij}$ and $\gamma_{ij}$ denote per-spring stiffness and damping parameters.

The optimization objective at each stage $k \in \{0, 1\}$ (where $k = 0$ and $k = 1$ correspond to the sparse and dense stages, respectively) can be written as:

$$\min_{\Theta_k} \ \frac{1}{T} \sum_{t=1}^{T} \mathcal{L}_{\text{chamfer}}(\hat{\mathbf{S}}_t, \mathbf{S}_t) + \lambda \mathcal{L}_{\text{track}}(\hat{\mathbf{S}}_t, \mathbf{S}_t) \quad \text{s.t.} \quad \hat{\mathbf{S}}_t = f(\hat{\mathbf{S}}_{t-1}, \Theta_0, \Theta_1, \Theta_{hand}), \ \hat{\mathbf{S}}_0 = \mathbf{S}_0, \ (5)$$

Here, $\hat{\mathbf{S}}_t$ denotes the simulated state, $\mathbf{S}_t$ denote the observed state from the inputs, and $f$ denote a simulation forward function based on the spring-mass system. As discussed in Sec. 3.2, $\mathcal{L}_{\text{chamfer}}$ measures the Chamfer distance to encourage simulated nodes to remain close to the 3D point cloud lifted from the input depth maps, while $\mathcal{L}_{\text{track}}$ measures the $\ell_2$ discrepancy to per-frame 3D tracked points obtained from CoTracker3 (Karaev et al., 2024).

For the sparse stage, we use zero-order optimization (Lozano, 2006) for 100 iterations, with initialization values of $\delta$, $d_{\max}$, and $s_{\text{global}}$ set to 0.002 and 3, respectively. For the dense stage, we use the Adam optimizer (Kingma, 2014) for 200 iterations with an initial learning rate of $1 \times 10^{-3}$. All other hyperparameters are kept identical to Jiang et al. (2025).

## C.3 Hand Refinement

In this stage, we refine the initial MANO parameters $\Theta_{hand}$ to produce object simulations better aligned with the input observations, using the spring–mass model fitted in the previous stage. An overview of this stage, including the loss function, is provided in Sec. 3.2. For the loss function in Eq. 2, we set $\lambda_{\text{track}} = 1$. Optimization is performed with the Adam optimizer (Kingma, 2014) with 40 optimization steps. The MANO parameters are initialized from the fitting results of the initial hand reconstruction stage, with an initial learning rate of $2 \times 10^{-5}$ decayed by 0.99 at each iteration.

## D Disclosure of LLM Usage

The writing of this manuscript was aided by LLMs, which were used to refine the initial draft for clarity and grammar. LLMs were not used for any other purpose.

## E Robustness Analysis

To evaluate the robustness of our method under imperfect upstream signals, we conducted a unified set of perturbation studies covering all components that influence object reconstruction—including depth input, CoTracker (Karaev et al., 2024) trajectories, and the MANO (Romero et al., 2017)-based hand controller. These perturbations reflect realistic deviations that can arise in sparse-view RGB-D capture, where even small discrepancies in depth or pixel tracking may lead to noticeable 3D

deviations due to geometric amplification. We first injected $\sim$1 mm perturbations into the depth maps and $\sim$1 px shifts into the CoTracker tracks. We additionally perturbed the MANO parameters such that the resulting hand pose exhibits a Mean Per-Joint Position Error (MPJPE) of $\sim$10 mm, consistent with the accuracy range of recent image-based MANO estimators (Zheng et al., 2023). For fairness, comparable perturbations were also applied to the sparse controller used in PhysTwin (Jiang et al., 2025).

Tab. 6 summarizes all results, where numbers in parentheses denote performance changes relative to the clean-input baseline. Across all perturbation types, PhysTwin (Jiang et al., 2025) shows a pronounced degradation—especially under tracking noise, where instability in its sparse control points leads to large reconstruction errors. In contrast, ours exhibits noticeably smaller performance drops. Our denser hand reconstruction provides more reliable contact cues and distributes the effect of upstream noise, resulting in only modest accuracy changes even when depth, tracks, or controller parameters are perturbed.

Overall, these observations indicate that our pipeline maintains stable reconstruction quality under realistic input noise across all upstream components, demonstrating significantly stronger robustness than PhysTwin (Jiang et al., 2025) under identical perturbation settings. While performance naturally degrades as noise increases, the overall robustness trends also suggest that the method could extend to settings where depth or hand-pose signals come from modern prediction models rather than optimized or ground-truth inputs, indicating a possible path toward broader use beyond controlled capture environments.

| Method | Dense interaction sequences | | Full sequences | |
|---|---|---|---|---|
| | CD ↓ | Track Err. ↓ | CD ↓ | Track Err. ↓ |
| PhysTwin Jiang et al. (2025) | 5.90 | 1.00 | 5.52 | 0.97 |
| **PHYSHANDI (Ours)** | **5.30** | **0.89** | **5.40** | **0.96** |
| PhysTwin (Noisy depth) | 6.93 (1.03) | 1.12 (0.12) | 6.34 (0.82) | 1.12 (0.15) |
| **PHYSHANDI (Noisy depth)** | **6.19 (0.89)** | **1.00 (0.11)** | **6.10 (0.70)** | **1.05 (0.09)** |
| PhysTwin (Noisy track) | 9.60 (3.70) | 1.46 (0.46) | 8.32 (2.80) | 1.34 (0.37) |
| **PHYSHANDI (Noisy track)** | **5.56 (0.26)** | **0.86 (-0.03)** | **5.50 (0.10)** | **0.92 (-0.04)** |
| PhysTwin (Noisy controller) | 7.54 (1.64) | 1.25 (0.25) | 6.89 (**1.37**) | **1.25 (0.28)** |
| **PHYSHANDI (Noisy controller)** | **6.44 (1.14)** | **1.08 (0.19)** | **6.79** (1.39) | 1.29 (0.33) |

Table 6: **Robustness analysis** under perturbations applied to depth input, CoTracker (Karaev et al., 2024) trajectories, and hand-pose controller parameters. We compare PhysTwin (Jiang et al., 2025) and our method under identical perturbation settings. Values in parentheses denote performance changes relative to the clean-input baseline.

