# OpenReview forum: "PhysHandi: Physics-Based Reconstruction of Hand-Deformable Object Interactions"
_ICLR.cc/2026/Conference — Submitted to ICLR 2026_

### Official Review · Reviewer_rJqj · 2025-10-15

**Soundness:** 2
**Presentation:** 2
**Contribution:** 2
**Rating:** 2
**Confidence:** 5

**Summary:**

This paper, PHYSHANDI: Physics-Based Reconstruction of Hand–Deformable Object Interactions, proposes a framework for dense 3D reconstruction of interacting hands and deformable objects. The key idea is to couple a parametric hand model (MANO) with a spring–mass simulation of deformable objects, where interaction forces are induced by hand mesh motion. The pipeline consists of three stages: hand reconstruction, object reconstruction, and hand refinement through inverse physics. Experiments on the PhysTwin dataset and a newly collected DENSEHDI dataset show improvements over the prior baseline PhysTwin in reconstruction, future prediction, and generalization

**Strengths:**

1. Inverse physics for hand refinement: The idea of improving hand reconstruction by leveraging reconstructed deformable object dynamics is interesting.

2. Dataset contribution: The authors additionally collect a new dataset (DENSEHDI) featuring denser hand–object contacts, which could benefit the community.

**Weaknesses:**

1. Limited generalization beyond lab conditions:
All experiments are conducted in highly controlled RGB-D capture setups (three synchronized RealSense cameras). There is no evidence that the method generalizes to in-the-wild scenarios (e.g., monocular RGB videos, unconstrained lighting, cluttered backgrounds). Given the reliance on multi-view depth, the method is unlikely to scale to real-world applications.

2. Strong assumption on contact regions:
As discussed in Section 5, the framework assumes a fixed hand–object contact topology within a sequence. This is a very strong and unrealistic assumption: in real interactions, contacts appear and disappear dynamically (e.g., fingers releasing or sliding over cloth). This limitation significantly reduces the applicability of the method.

3. Lack of extensive visualizations:
While a few qualitative figures are shown in the main paper (e.g., Fig. 3), the visual results are limited in scope. For a reconstruction method, more extensive qualitative evidence (especially videos in supplementary material) is crucial for evaluating realism and stability. Without this, it is hard to be fully convinced of the claimed improvements.

**Questions:**

Please refer to Weaknesses.

---

> ### Author Response · Authors · 2025-11-26
>
> We thank you for your valuable comments, which provide helpful insights to improve our manuscript. We will incorporate the minor suggestions in the revision. Below, we address the main concerns raised in the review.
>
> ---
>
> > **Weakness 1.** "Limited generalization beyond lab conditions: All experiments are conducted in highly controlled RGB-D capture setups (three synchronized RealSense cameras). There is no evidence that the method generalizes to in-the-wild scenarios (e.g., monocular RGB videos, unconstrained lighting, cluttered backgrounds). Given the reliance on multi-view depth, the method is unlikely to scale to real-world applications."
>
> **Reply:** Our target problem is indeed 3D hand–deformable-object model fitting in a controlled multi-view RGB-D setup, *which we directly adopted from recent prior work (PhysTwin) to enable fair comparisons*. We would like to note that such 3D reconstruction from controlled multi-view RGB-D data (e.g., [Hampali et al., 2020; Xie et al., 2023; Cho et al., 2024]) still remains an active research problem; nevertheless, we also demonstrated results under the more challenging monocular RGB-D setting in Tab. 3 of the paper.
>
> Additionally, we also indirectly validate that our method can partially relax the controlled settings via evaluating the robustness of our system under noisy depth, noisy tracks, and degraded hand estimates in Tab. R1. Specifically, we add ~1 mm noise to the input depth, ~1 px perturbation to the CoTracker tracks, and perturbations to the MANO parameters such that the resulting hand pose has an MPJPE of ~10 mm. We believe these results specifically highlight the potential of our method to scale to real-world applications based on monocular or multi-view RGB-only videos -- where our hand–deformable-object model would be fitted to the predicted 3D surface, depth, or hand estimates based on such RGB inputs -- as our simulated noise levels remain within the accuracy range of the current state-of-the-art estimators. For example, sparse few-shot 3D surface reconstruction approaches such as NeuSurf and FatesGS achieve Chamfer distances of 1.35 and 1.37, respectively. Likewise, the hand-parameter noise level we simulate aligns with the accuracy of state-of-the-art MANO estimators operating on single frames [R1]. Following your suggestion, we will gradually scale our framework to real-world applications in future work.
>
> **Table R1**
>
> | **Method**                       | **CD ↓ (Dense Seq.)**    | **Track Err. ↓ (Dense Seq.)** | **CD ↓ (Full Seq.)**     | **Track Err. ↓ (Full Seq.)** |
> | -------------------------------- | ------------------- | ------------------------ | ------------------- | ----------------------- |
> | PhysTwin (Jiang et al. 2025)     | 5.90                | 1.00                     | 5.52                | 0.97                    |
> | **PhysHandi (Ours)**             | **5.30**            | **0.89**                 | **5.40**            | **0.96**                |
> |                                  |                     |                          |                     |                         |
> | PhysTwin (Noisy depth)           | 6.93 (1.03)         | 1.12 (0.12)              | 6.34 (0.82)         | 1.12 (0.15)             |
> | **PhysHandi (Noisy depth)**      | **6.19** (**0.89**) | **1.00** (**0.11**)      | **6.10** (**0.70**) | **1.05** (**0.09**)     |
> |                                  |                     |                          |                     |                         |
> | PhysTwin (Noisy track)           | 9.60 (3.70)         | 1.46 (0.46)              | 8.32 (2.80)         | 1.34 (0.37)             |
> | **PhysHandi (Noisy track)**      | **5.56** (**0.26**) | **0.86** (**-0.03**)     | **5.50** (**0.10**) | **0.92** (**-0.04**)    |
> |                                  |                     |                          |                     |                         |
> | PhysTwin (Noisy controller)      | 7.54 (1.64)         | 1.25 (0.25)              | 6.89 (**1.37**)     | **1.25** (**0.28**)     |
> | **PhysHandi (Noisy controller)** | **6.44** (**1.14**) | **1.08** (**0.19**)      | **6.79** (1.39)     | 1.29 (0.33)             |
>
> [R1] Zheng et. al., HaMuCo: Hand Pose Estimation via Multiview Collaborative Self-Supervised Learning, In ICCV, 2023.

---

> > ### Author Response · Authors · 2025-11-26
> >
> > > **Weakness 2.** "Strong assumption on contact regions: As discussed in Section 5, the framework assumes a fixed hand–object contact topology within a sequence. This is a very strong and unrealistic assumption: in real interactions, contacts appear and disappear dynamically (e.g., fingers releasing or sliding over cloth). This limitation significantly reduces the applicability of the method."
> >
> > **Reply:** While we agree that the assumption of a fixed hand–object contact topology per sequence is a limitation (as discussed in Sec. 5), we would like to respectfully note that this assumption is common in existing multi-view hand–rigid-object fitting methods (e.g., [Hampali et al., 2020; Xie et al., 2023; Cho et al., 2024]) as well as in deformable-object fitting methods such as PhysTwin [Jiang et al., 2025], where it is used to keep the optimization problem tractable. While dynamic contact handling remains a shared limitation across these existing works, following your suggestion, we will try to further investigate addressing this limitation in future work.
> >
> > At the same time, we would like to highlight that our method still represents a meaningful advancement over the prior state of the art, as it is the ***first*** to tackle full 3D reconstruction of both hands and objects undergoing large-scale non-rigid deformations – a more challenging task than those those addressed in existing prior work (as mentioned in Sec. 2 in the paper). Toward this goal, our method effectively models dense hand-deformable-object interactions, as shown in the  supplementary video and the quantitative comparisons updated in the revised manuscript. We sincerely hope that our improvements over prior work can be taken into consideration during the evaluation.
> >
> > ---
> >
> > > **Weakness 3.** "Lack of extensive visualizations: While a few qualitative figures are shown in the main paper (e.g., Fig. 3), the visual results are limited in scope. For a reconstruction method, more extensive qualitative evidence (especially videos in supplementary material) is crucial for evaluating realism and stability. Without this, it is hard to be fully convinced of the claimed improvements."
> >
> > **Reply:** We thank the reviewer for the thoughtful comment. Following your suggestion, we provide more extensive qualitative comparison results on the PhysTwin and DenseHDI datasets in the supplementary video. As shown in the video, PhysHanDI significantly outperforms all baselines, particularly in handling highly dense hand–deformable-object interactions. We will include these additional qualitative comparisons and discussions in the revision.

---

### Official Review · Reviewer_S4t5 · 2025-10-20

**Soundness:** 2
**Presentation:** 3
**Contribution:** 2
**Rating:** 4
**Confidence:** 4

**Summary:**

This paper introduces PHYSHANDI, a physics-based framework for reconstructing hand-deformable object interactions from sparse-view RGB-D videos. PHYSHANDI reconstructs deformable objects with the interaction forces, which are simulated based on reconstructed hand motions. The method further refines hand reconstruction using inverse physics from object deformations.

**Strengths:**

1. The paper is well-written and well-organized.

2. The topic of hand-deformable object reconstruction is interesting.

**Weaknesses:**

1. More HOI methods using Spring-Mass models, such as CPF [1], have not been discussed.

2. The proposed framework relies on the initial MANO-based hand reconstruction to drive the subsequent deformable object simulation. If the hand mesh contains significant errors or inaccuracies, to what extent can PHYSHANDI still reconstruct a meaningful deformable object? Have the authors evaluated the robustness of the object reconstruction under degraded or noisy hand estimates?

3. Is the proposed method sufficiently innovative compared to the baseline method? For example, is there any substantial difference between Section 3.1 and PhysTwin?

4. The paper only compares one method. Please provide comparisons with more methods on more datasets. For the qualitative comparison, please provide results from multiple methods on the same test samples.

[1] CPF: Learning a Contact Potential Field to Model the Hand-Object Interaction. ICCV 2021.

**Questions:**

Same as the above weaknesses.

---

> ### Author Response · Authors · 2025-11-25
>
> We thank you for your valuable comments, which provide helpful insights to improve our manuscript. We will incorporate the minor suggestions in the revision. Below, we address the main concerns raised in the review.
>
> ---
>
> >**Weakness 1.** "More HOI methods using Spring-Mass models, such as CPF [1], have not been discussed."
>
> **Reply:** We thank the reviewer for the suggestion and will include CPF in the related work section. In a nutshell, we believe that CPF's problem scope and use of a spring–mass system differ from ours: in CPF, objects are assumed to be rigid and no object deformation is simulated, and the springs are used only locally in the direct hand–object contact region to regularize hand and **rigid-object poses**. In contrast, ours employs a full spring–mass simulation to model the entire object's **deformation** dynamics, which in turn drives and refines both object and hand reconstruction via inverse physics. As our work primarily targets such deformable object–hand interaction modeling, we initially did not prioritize discussing this method due to its different problem scope, but we agree that adding it will help clarify this distinction in the revision. We again thank the reviewer for the helpful suggestion.
>
> ---
>
> >**Weakness 2.** "The proposed framework relies on the initial MANO-based hand reconstruction to drive the subsequent deformable object simulation. If the hand mesh contains significant errors or inaccuracies, to what extent can PHYSHANDI still reconstruct a meaningful deformable object? Have the authors evaluated the robustness of the object reconstruction under degraded or noisy hand estimates?"
>
> **Reply:** Thank you for the question. Following your suggestion, we additionally evaluated the robustness of our system under degraded hand estimates, where we injected perturbations into the MANO parameters so that the resulting hand pose has an MPJPE of ~10 mm. For comparisons, we applied noise of comparable magnitude to PhysTwin's sparse controller as well and evaluated the object reconstruction accuracy.
>
> As shown in Tab. R1, our method mostly outperforms the baseline, demonstrating the effectiveness under this perturbed setting as well - especially for modeling dense hand-deformable-object interactions. We will include these experimental results and the accompanying discussion in the revision.
>
> **Table R1**
>
> | **Method**                       | **CD ↓** (Dense Seq.)    | **Track Err. ↓** (Dense Seq.) | **CD ↓** (Full Seq.)     | **Track Err. ↓** (Full Seq.) |
> | -------------------------------- | ------------------- | ------------------------ | ------------------- | ----------------------- |
> | PhysTwin (Noisy controller)      | 7.54    | 1.25          | 6.89    | **1.25**    |
> | **PhysHandi (Noisy controller)** | **6.44**  | **1.08**      | **6.79**     | 1.29           |
>
> ---

---

> ### Author Response · Authors · 2025-11-25
>
> >**Weakness 3.** "Is the proposed method sufficiently innovative compared to the baseline method? For example, is there any substantial difference between Section 3.1 and PhysTwin?"
>
> **Reply:** Although our deformable-object model fitting procedure is based on PhysTwin (L254–256), we would like to highlight that our core contribution is the integration of hand reconstruction and deformable-object reconstruction into a unified, physics-based loop that enables bidirectional enhancement: more accurate hand estimates improve object simulation, and improved object simulation in turn enhances hand fitting. This bidirectional synergy (hand modeling ↔ object modeling) goes beyond the "deformable-object-only model fitting" scope of PhysTwin.
>
> Thanks to these contributions, PhysHanDI is noticeably more effective in modeling highly dense hand–deformable-object interactions than the prior state-of-the-art. To support this, we provide additional qualitative comparisons in the supplementary video, and also report quantitative comparisons using an additional metric that measures Chamfer distance only for points undergoing actual deformation (CD_dyn). Note that the original 3D shape accuracy metric (CD_org) used by PhysTwin computes distances over both deforming and static points, and therefore may be less effective for capturing reconstruction accuracy for the deformable regions. The numerical gaps in CD_dyn clearly demonstrate the advantages of our method over PhysTwin.
>
> **Table R2**
>
> **(a) Results on the dense interaction sequences**
>
> | **Method**                 | **CD_org ↓** | **CD_dyn ↓** | **CD_org ↓ (Future Prediction)** | **CD_dyn ↓ (Future Prediction)** |
> |----------------------------|---------------|---------------|--------------------------|--------------------------|
> | PhysTwin (Jiang et al., 2025) | 5.90          | 10.78         | 11.45                   | 16.32                    |
> | **PhysHandi (Ours)**         | **5.30**      | **8.32**      | **10.57**               | **14.35**                |
>
>
> **(b) Results on the full sequences**
>
> | **Method**                 | **CD_org ↓** | **CD_dyn ↓** | **CD_org ↓ (Future Prediction)** | **CD_dyn ↓ (Future Prediction)** |
> |----------------------------|---------------|---------------|--------------------------|--------------------------|
> | PhysTwin (Jiang et al., 2025) | 5.52          | 7.63          | 12.26                   | 14.42                    |
> | **PhysHandi (Ours)**         | **5.40**      | **7.30**      | **12.04**               | **13.63**                |

---

> ### Author Response · Authors · 2025-11-25
>
> >**Weakness 4.** "The paper only compares one method. Please provide comparisons with more methods on more datasets. For the qualitative comparison, please provide results from multiple methods on the same test samples."
>
> **Reply:** Thank you for the suggestion. While the main paper focused on comparisons with the current state-of-the-art, PhysTwin, we would like to kindly clarify that an additional baseline (Spring-Gaus) was included in Sec. C of the supplementary. To further strengthen our evaluation, we also conducted new experiments with GS-Dynamics in this rebuttal. As shown in Tab. R3 below, PhysHanDI outperforms all baselines.
>
> Regarding the datasets, PhysTwin was the only existing benchmark for reconstructing hand–deformable-object interactions. To address this limitation, we newly collected the DenseHDI dataset, which contains more dense hand-object interactions, and additionally performed our experiments on it (Tabs. 1–2 in the paper).
>
> Finally, following the reviewer's request, we provide qualitative comparisons "from multiple methods on the same test samples" in the supplementary video. These results consistently show that PhysHanDI more robustly and accurately models highly dense hand–deformable-object interactions compared to all baselines. We thank your feedback and have included these additional results in the paper.
>
> **Table R3**
>
> **(a) Results on the dense interaction sequences**
>
> [Reconstruction & Resimulation]
>
> | **Method**                       | **CD_org ↓** | **CD_dyn ↓** | **Track Err. ↓** | **IoU ↑** | **PSNR ↑** | **SSIM ↑** | **LPIPS ↓** |
> | -------------------------------- | ------------- | ------------ | ---------------- | --------- | ---------- | ---------- | ----------- |
> | Spring-Gaus (Zhong et al. 2024)  | 38.84         | 27.79        | 4.65             | 0.55      | 21.41      | 0.94       | 7.80        |
> | GS-Dynamics (Zhang et al. 2024a) | 13.67         | 33.37        | 1.81             | 0.74      | 23.12      | 0.94       | 3.80        |
> | PhysTwin (Jiang et al. 2025)     | 5.90          | 10.78        | 1.00             | 0.84      | 25.23      | **0.95**   | 2.65        |
> | **PhysHandi (Ours)**             | **5.30**      | **8.32**     | **0.89**         | **0.85**  | **25.62**  | **0.95**   | **2.52**    |
>
> [Future Prediction]
>
> | **Method**                       | **CD_org ↓** | **CD_dyn ↓** | **Track Err. ↓** | **IoU ↑** | **PSNR ↑** | **SSIM ↑** | **LPIPS ↓** |
> | -------------------------------- | ------------- | ------------ | ---------------- | --------- | ---------- | ---------- | ----------- |
> | Spring-Gaus (Zhong et al. 2024)  | 56.51         | 37.38        | 7.69             | 0.42      | 19.91      | **0.94**   | 9.27        |
> | GS-Dynamics (Zhang et al. 2024a) | 33.99         | 56.79        | 4.50             | 0.51      | 18.97      | 0.93       | 7.79        |
> | PhysTwin (Jiang et al. 2025)     | 11.45         | 16.32        | 2.10             | 0.70      | 22.07      | **0.94**   | 5.38        |
> | **PhysHandi (Ours)**             | **10.57**     | **14.35**    | **2.05**         | **0.73**  | **22.84**  | **0.94**   | **4.83**    |
>
> **(b) Results on the full sequences**
>
> [Reconstruction & Resimulation]
>
> | **Method**                       | **CD_org ↓** | **CD_dyn ↓** | **Track Err. ↓** | **IoU ↑** | **PSNR ↑** | **SSIM ↑** | **LPIPS ↓** |
> | -------------------------------- | ------------- | ------------ | ---------------- | --------- | ---------- | ---------- | ----------- |
> | Spring-Gaus (Zhong et al. 2024)  | 33.60         | 26.39        | 4.07             | 0.62      | 21.24      | 0.94       | 8.09        |
> | GS-Dynamics (Zhang et al. 2024a) | 13.79         | 24.73        | 2.18             | 0.72      | 24.01      | **0.95**   | 4.14        |
> | PhysTwin (Jiang et al. 2025)     | 5.52          | 7.63         | 0.97             | **0.84**  | 26.32      | **0.95**   | 2.88        |
> | **PhysHandi (Ours)**             | **5.40**      | **7.30**     | **0.96**         | **0.84**  | **26.44**  | **0.95**   | **2.87**    |
>
> [Future Prediction]
>
> | **Method**                       | **CD_org ↓** | **CD_dyn ↓** | **Track Err. ↓** | **IoU ↑** | **PSNR ↑** | **SSIM ↑** | **LPIPS ↓** |
> | -------------------------------- | ------------- | ------------ | ---------------- | --------- | ---------- | ---------- | ----------- |
> | Spring-Gaus (Zhong et al. 2024)  | 46.54         | 49.29        | 6.61             | 0.48      | 19.59      | 0.93       | 9.44        |
> | GS-Dynamics (Zhang et al. 2024a) | 38.84         | 52.96        | 6.88             | 0.46      | 19.38      | 0.94       | 8.32        |
> | PhysTwin (Jiang et al. 2025)     | 12.26         | 14.42        | 2.44             | **0.69**  | 22.80      | **0.95**   | 5.15        |
> | **PhysHandi (Ours)**             | **12.04**     | **13.63**    | **2.41**         | 0.68      | **22.96**  | **0.95**   | **5.02**    |

---

### Official Review · Reviewer_asRB · 2025-10-31

**Soundness:** 2
**Presentation:** 3
**Contribution:** 2
**Rating:** 6
**Confidence:** 4

**Summary:**

The paper proposes PHYSHANDI, a physics-based framework that jointly reconstructs a full 3D hand and deformable objects from sparse RGB-D sequences. The method treats the dense MANO hand mesh as boundary conditions that drive a spring-mass object via virtual springs, and then performs inverse-physics refinement so that the learned object model provides supervision and correction for the hand. Experiments on PhysTwin scenarios and the new DENSEHDI dataset report improvements over PhysTwin on reconstruction/resimulation, future prediction, and generalization.

**Strengths:**

1. The work specifically targets deformable-object interaction within hand–object reconstruction, which remains underexplored, and it fills a tangible gap.
2. The supp. discusses the choice of $\delta$, a key hyperparameter; surfacing part of this analysis in the main paper would further clarify the design.
3. The paper releases a dataset that benefits the community.

**Weaknesses:**

1. Relative to PhysTwin, the contribution appears incremental. The main change introduces MANO as a topological prior for the hand to improve physical modeling near contact. This design understandably differs from PhysTwin, which targets a robot-arm-plus-gripper setting and therefore does not emphasize detailed hand modeling.
2. MANO has limited expressiveness for elastic skin and soft-tissue effects. While it encodes hand topology, optimizing $\Theta$ alone does not capture elastic deformation.
3. The method depends on several potentially brittle components: accurate depth (which consumer sensors like D455 often noise), and CoTracker for initialization. The accuracy of CoTracker likely has a first-order effect on downstream object reconstruction quality.

**Questions:**

1. Beyond introducing MANO as a hand prior, does the approach support broader application scenarios than PhysTwin, or does it offer a deeper theoretical contribution that the paper can articulate more explicitly?
2. When modeling hand deformation, do the authors consider alternatives (e.g., non-rigid extensions, blendshape or learned corrective fields) that may better account for elastic effects than optimizing $\Theta$ alone?
3. Can the authors include ablations on input quality (e.g., depth noise, missing data) and on dependencies (e.g., CoTracker accuracy, initialization errors) to quantify robustness?

---

> ### Author Response · Authors · 2025-11-26
>
> We thank you for your valuable comments, which provide helpful insights to improve our manuscript. We will incorporate the minor suggestions in the revision. Below, we address the main concerns raised in the review.
>
> ---
>
> > **Weakness 1 / Question 1.** "Relative to PhysTwin, the contribution appears incremental. The main change introduces MANO as a topological prior for the hand to improve physical modeling near contact. This design understandably differs from PhysTwin, which targets a robot-arm-plus-gripper setting and therefore does not emphasize detailed hand modeling."
>
> **Reply:** We thank the reviewer for the thoughtful comment. As noted, PhysTwin does not target full 3D hand-shape reconstruction (instead directly using the input depth map to represent partial hand geometry), whereas PhysHanDI takes a first step toward joint full 3D reconstruction of both interacting hands and non-rigid objects by integrating both tasks within a unified, physics-based loop. This loop enables synergistic improvement of both components: more accurate hand estimates lead to better object simulation, and improved object simulation in turn enhances hand fitting. In Sec. A of the supplementary, we also analyzed how our MANO-based hand representation yields a more effective spring–mass model topology, resulting in more accurate physical simulation.
>
> Thanks to these contributions, PhysHanDI noticeably outperforms PhysTwin, especially in modeling highly dense hand–deformable-object interactions. To further support this, we provide additional qualitative comparisons in the supplementary video, and also report quantitative comparisons using an additional metric that measures Chamfer distance only for points undergoing actual deformation (CD_dyn). Note that the original 3D shape accuracy metric (CD_org) used by PhysTwin computes distances over both deforming and static points, and therefore may be less effective in capturing reconstruction accuracy for the deformable regions. The numerical gaps in CD_dyn clearly demonstrate the advantages of PhysHanDI over PhysTwin.
>
> Additionally, we believe that PhysHanDI also provides important benefits for the robustness of simulation-based reconstruction. As shown in Tab. 3 of the paper, PhysTwin’s simulation fails in more challenging single-view RGB-D setups due to its difficulty in modeling close hand–object contacts while relying on sparser input depth. In contrast, PhysHanDI performs robust simulation thanks to full 3D hand-shape modeling based on MANO. Later in the author response (for Weakness 3), we also empirically demonstrate that PhysHanDI remains noticeably more robust when the input signals (e.g., depth or 2D tracking) are noisy, again due to the full 3D hand-shape modeling.
>
> **Table R1**
>
> **(a) Results on the dense interaction sequences**
>
> | **Method**                    | **CD_org ↓** | **CD_dyn ↓** | **CD_org ↓ (Future Prediction)** | **CD_dyn ↓ (Future Prediction)** |
> | ----------------------------- | ------------- | ------------ | ---------------------- | ---------------------- |
> | PhysTwin (Jiang et al., 2025) | 5.90          | 10.78        | 11.45                  | 16.32                  |
> | **PhysHandi (Ours)**          | **5.30**      | **8.32**     | **10.57**              | **14.35**              |
>
> **(b) Results on the full sequences**
>
> | **Method**                    | **CD_org ↓** | **CD_dyn ↓** | **CD_org ↓ (Future Prediction)** | **CD_dyn ↓ (Future Prediction)** |
> | ----------------------------- | ------------- | ------------ | ---------------------- | ---------------------- |
> | PhysTwin (Jiang et al., 2025) | 5.52          | 7.63         | 12.26                  | 14.42                  |
> | **PhysHandi (Ours)**          | **5.40**      | **7.30**     | **12.04**              | **13.63**              |
>
> ---

---

> ### Author Response · Authors · 2025-11-26
>
> > **Weakness 2 / Question 2.** "MANO has limited expressiveness for elastic skin and soft-tissue effects. While it encodes hand topology, optimizing alone does not capture elastic deformation."
>
> **Reply:** We appreciate the reviewer’s insightful comment. We indeed adopted a relatively simple MANO representation for hands, which has limited expressiveness for modeling elastic skin and soft-tissue effects. While this is a current limitation, we would like to kindly note that our target problem is still more challenging than those addressed in existing prior work. Our method represents a first step toward joint full 3D reconstruction of hands and highly deformable objects, whereas previous work focuses only on modeling (1) partial / sparse controller and deformable object interactions (PhysTwin [Jiang et al., 2025]) or (2) MANO hands and rigid or only locally deformable object interactions [Hampali et al., 2020; Cho et al., 2024; Xie et al., 2023], as discussed in the related work section.
>
> Thus, rather than focusing on more expressive hand modeling, we prioritized collecting a new dataset (DenseHDI) to capture dense and dynamic hand–deformable-object interactions, and demonstrating the effectiveness of the bidirectional synergy between MANO-based hand reconstruction and deformable-object reconstruction. Nevertheless, we agree that modeling more expressive hand shapes is an important future direction, and we will clarify this limitation more explicitly in the revised manuscript. Following your comment, we will explore, as future work, augmenting MANO with per-vertex displacement fields or a learned nonlinear deformation basis to better capture elastic deformations. We again thank the reviewer for the valuable feedback.
>
> ---
>
> > **Weakness 3 / Question 3.** "The method depends on several potentially brittle components: accurate depth (which consumer sensors like D455 often noise), and CoTracker for initialization. The accuracy of CoTracker likely has a first-order effect on downstream object reconstruction quality."
>
> **Reply:** Thank you for the valuable comment. To address your concern, we conducted additional experiments to evaluate the sensitivity of these components. Specifically, we compared the performance of our method and PhysTwin under noisy depth, noisy tracks, and degraded hand estimates. Specifically, we add ~1 mm noise to the input depth, ~1 px perturbation to the CoTracker tracks, and perturbations to the MANO parameters such that the resulting hand pose has an MPJPE of ~10 mm, respectively.
>
> As shown in Tab. R2, PhysTwin exhibits substantial performance degradation under these perturbations, particularly with noisy CoTracker inputs (note that numbers in parentheses denote performance changes relative to the original results). In contrast, PhysHanDI remains comparatively more robust across both noisy sources, as its denser hand reconstruction provides more reliable contact cues. Following your comment, we will include these experimental results in the revision.
>
> **Table R2**
>
> | **Method**                       | **CD ↓ (Dense Seq.)**    | **Track Err. ↓ (Dense Seq.)** | **CD ↓ (Full Seq.)**     | **Track Err. ↓ (Full Seq.)** |
> | -------------------------------- | ------------------- | ------------------------ | ------------------- | ----------------------- |
> | PhysTwin (Jiang et al. 2025)     | 5.90                | 1.00                     | 5.52                | 0.97                    |
> | **PhysHandi (Ours)**             | **5.30**            | **0.89**                 | **5.40**            | **0.96**                |
> |                                  |                     |                          |                     |                         |
> | PhysTwin (Noisy depth)           | 6.93 (1.03)         | 1.12 (0.12)              | 6.34 (0.82)         | 1.12 (0.15)             |
> | **PhysHandi (Noisy depth)**      | **6.19** (**0.89**) | **1.00** (**0.11**)      | **6.10** (**0.70**) | **1.05** (**0.09**)     |
> |                                  |                     |                          |                     |                         |
> | PhysTwin (Noisy track)           | 9.60 (3.70)         | 1.46 (0.46)              | 8.32 (2.80)         | 1.34 (0.37)             |
> | **PhysHandi (Noisy track)**      | **5.56** (**0.26**) | **0.86** (**-0.03**)     | **5.50** (**0.10**) | **0.92** (**-0.04**)    |
> |                                  |                     |                          |                     |                         |
> | PhysTwin (Noisy controller)      | 7.54 (1.64)         | 1.25 (0.25)              | 6.89 (**1.37**)     | **1.25** (**0.28**)     |
> | **PhysHandi (Noisy controller)** | **6.44** (**1.14**) | **1.08** (**0.19**)      | **6.79** (1.39)     | 1.29 (0.33)             |

---

### Official Review · Reviewer_rNHX · 2025-11-01

**Soundness:** 2
**Presentation:** 2
**Contribution:** 2
**Rating:** 4
**Confidence:** 3

**Summary:**

The paper introduces PHYSHANDI, a physics-based framework for reconstructing and simulating 3D hand-deformable object interactions from sparse-view RGB-D videos. The technical contributions include: (1) Dense hand modeling using the MANO parametric hand model. (2) Deformable object simulation via a spring-mass system, where object deformations are driven by forces from reconstructed hand motions. (3) A three-stage optimization pipeline. In experimetns, the method outperforms the state-of-the-art baseline (PhysTwin) in reconstruction accuracy, future prediction, and generalization to unseen interactions, particularly in scenarios with dense hand-object contacts.

**Strengths:**

1.	The novel dataset DENSEHDI focuses on dense hand-object contacts, addressing a gap in existing benchmarks.

2.	This paper achieves the dense 3D reconstruction of both hands and deformable objects simultaneously, ensuring physical plausibility and coherence between hands and object dynamics.

3.	The inverse physics design help improve accuracy in sparse-view settings, especially the single-view cases.

**Weaknesses:**

1.	Physics-based simulation and optimization may require significant computational resources. Computational cost should be clarified.

2.	This paper lacks experiments directly evaluating the accuracy of hand reconstructions. The title of the paper gives hand and object the equal position, but only evaluate objects. The claim in the Abstract “such simulation of object deformations can, in turn, refine and improve hand reconstruction via inverse physics” is not fully supported.

3.	The quantitative results do not significantly outperform previous PhysTwin method in Table. 1, especially for  metrics such as SSIM. Why?

4. It seems that the model assumes fixed hand-object contact topology within a sequence, limiting its applicability to interactions with dynamic contact changes (e.g., sliding or rolling).

5. Previous paper "InteractionFusion: Real-time Reconstruction of Hand Poses and Deformable Objects in Hand-object Interactions, ACM SIGGRAPH 2019." should be cited. This is "another kind" of hand-deformable object interactions". In comparison with this paper, this manuscript is more like a dynamic object simulation or cloth simulation paper with a hand-refinement module, which targets refining the object dynamics at the end.

**Questions:**

The main flaw is lack of experiments on hand reconstruction quality.

---

> ### Author Response · Authors · 2025-11-25
>
> We thank you for your valuable comments, which provide helpful insights to improve our manuscript. We will incorporate the minor suggestions in the revision. Below, we address the main concerns raised in the review.
>
> ---
>
> >**Weakness 1.** "Physics-based simulation and optimization may require significant computational resources. Computational cost should be clarified."
>
> **Reply:** Thank you for the comment. Following your suggestion, we additionally provide a detailed runtime breakdown of our method and also PhysTwin in Tab. R1. Since the simulation cost scales with sequence length, we report average per-frame runtimes in seconds.
> As shown in the table, our training and inference stages run faster than PhysTwin. This improvement directly stems from *“PHYSHANDI’s denser hand reconstruction, which localizes contact regions more precisely” (L781-782)* and therefore activates fewer springs during simulation. As visualized in Fig. 4 in the paper, our springs remain concentrated around correct contact areas, reducing per-spring force computation and explaining the observed speedup.
>
> **Table R1**
>
> | **Method**                     | **Training (Zero-order)** | **Training (First-order)** | **Inference (Simulation)** |
> |-------------------------------|----------------------------|-----------------------------|-----------------------------|
> | PhysTwin (Jiang et al., 2025) | 13.80                      | 21.22                       | 0.14                        |
> | **PhysHandi (Ours)**          | **12.24**                  | **17.63**                   | **0.11**                    |
>
> ---
>
> >**Weakness 2.** "This paper lacks experiments directly evaluating the accuracy of hand reconstructions. The title of the paper gives hand and object the equal position, but only evaluate objects. The claim in the Abstract “such simulation of object deformations can, in turn, refine and improve hand reconstruction via inverse physics” is not fully supported."
>
> **Reply:** We would like to kindly clarify that hand fitting accuracy is directly evaluated in Tab. 3 in the paper (Hand CD metric) for the single-view hand prediction setting. This experiment was designed to validate our claim that *“simulation of object deformations can, in turn, refine and improve hand reconstruction via inverse physics,”* by showing that inverse-physics-based hand refinement improves this hand accuracy metric compared to its counterpart. We also highlight that, under this single-view setup, the existing baseline (PhysTwin) encounters simulation failures (L428–431), whereas our method robustly simulates hand–deformable-object interactions thanks to the full 3D hand shape modeling based on the MANO model.
>
> Nevertheless, as discussed in Sec. 5 in the paper, we acknowledge that evaluating multi-view (cf. single-view) hand reconstruction accuracy would be also valuable. However, as discussed (L510–519), the direct evaluation in this setting is not feasible due to the lack of ground-truth 3D hand labels beyond the input multi-view RGB-D frames. Indeed, existing works treat “MANO fitting to multi-view RGB-D” (L513) as a procedure to annotate ground-truth 3D hand shapes in datasets lacking such labels (L514–515). Following such multi-view-based methods [Hampali et al., 2020; Xie et al., 2023; Cho et al., 2024] that indirectly assess the the MANO fitting quality via downstream applications (e.g., single-view hand reconstruction), we also evaluate hand fitting quality through our downstream tasks: (1) physics-based deformable object reconstruction, where improvements in hand estimation directly translate to more accurate object simulations (L515–517), and (2) single-view hand prediction (Tab. 3) by regarding the multi-view RGB-D data as the ground truth. Although this was briefly discussed in Sec. 5 in the paper, we will further clarify this in the revision.
>
> ---

---

> ### Author Response · Authors · 2025-11-25
>
> >**Weakness 3.** “The quantitative results do not significantly outperform previous PhysTwin method in Table. 1, especially for metrics such as SSIM. Why?”
>
> **Reply:** We agree that the numerical performance gap over PhysTwin is not large on some metrics. However, we would like to note that this is somewhat influenced by the current metric design used in the baseline work (Jiang et al., 2025), which measures 3D or 2D accuracy over both deforming and non-deforming (static) object regions. In some sequences, the non-deforming region is larger than the deforming region, and this static region will always produce low error regardless of the actual dynamic deformable-shape reconstruction performance. This can offset the performance gap in the quantitative metrics.
>
> To address this, we report an additional metric—Chamfer distance computed only over points undergoing actual deformation (CD_dyn)—in Tab. R2 below. The numerical gaps in CD_dyn clearly demonstrate the advantages of our method over PhysTwin. We also provide additional qualitative comparisons on the DenseHDI dataset in the supplementary video, where our method achieves notably more robust and accurate reconstruction, especially for highly dense hand–deformable-object interactions.
>
> **Table R2**
>
> **(a) Results on the dense interaction sequences**
>
> | **Method**                 | **CD_org ↓** | **CD_dyn ↓** | **CD_org↓ (Future Prediction)** | **CD_dyn ↓ (Future Prediction)** |
> |----------------------------|---------------|---------------|--------------------------|--------------------------|
> | PhysTwin (Jiang et al., 2025) | 5.90          | 10.78         | 11.45                   | 16.32                    |
> | **PhysHandi (Ours)**         | **5.30**      | **8.32**      | **10.57**               | **14.35**                |
>
>
> **(b) Results on the full sequences**
>
> | **Method**                 | **CD_org ↓** | **CD_dyn ↓** | **CD_org ↓ (Future Prediction)** | **CD_dyn ↓ (Future Prediction)** |
> |----------------------------|---------------|---------------|--------------------------|--------------------------|
> | PhysTwin (Jiang et al., 2025) | 5.52          | 7.63          | 12.26                   | 14.42                    |
> | **PhysHandi (Ours)**         | **5.40**      | **7.30**      | **12.04**               | **13.63**                |
>
> ---
>
> >**Weakness 4.** “It seems that the model assumes fixed hand-object contact topology within a sequence, limiting its applicability to interactions with dynamic contact changes (e.g., sliding or rolling).”
>
> **Reply:** While we agree that the assumption of a fixed hand–object contact topology per sequence is a limitation (as discussed in Sec. 5), we would like to respectfully note that this assumption is common in existing multi-view hand–rigid-object fitting methods (e.g., [Hampali et al., 2020; Xie et al., 2023; Cho et al., 2024]) as well as in deformable-object fitting methods such as PhysTwin [Jiang et al., 2025], where it is used to keep the optimization problem tractable. While dynamic contact handling remains a shared limitation across these existing works, following your suggestion, we will try to further investigate addressing this limitation in future work.
>
> At the same time, we would like to highlight that our method still represents a meaningful advancement over the prior state of the art, as it is the ***first*** to tackle full 3D reconstruction of both hands and objects undergoing large-scale non-rigid deformations – a more challenging task than those addressed in existing prior work (as mentioned in Sec. 2 in the paper). We sincerely hope that our improvements over prior work can be taken into consideration during the evaluation.
>
> ---

---

> ### Author Response · Authors · 2025-11-25
>
> >**Weakness 5.** "Previous paper "InteractionFusion: Real-time Reconstruction of Hand Poses and Deformable Objects in Hand-object Interactions, ACM SIGGRAPH 2019." should be cited. This is "another kind" of hand-deformable object interactions". In comparison with this paper, this manuscript is more like a dynamic object simulation or cloth simulation paper with a hand-refinement module, which targets refining the object dynamics at the end.”"
>
> **Reply:** Thank you for suggesting this relevant prior work; we will include a discussion of InteractionFusion in the related work section. In a nutshell, we believe the key difference between this work and ours is that InteractionFusion is limited to modeling ***local*** object deformations induced by hand–object contact pressure—similar to the few existing hand–deformable-object interaction modeling methods (Xie et al., 2023; Qi et al., 2025) we discuss in the paper (L45–49). InteractionFusion explicitly notes that “non-rigidity does not happen everywhere; it majorly happens around contact points where external forces may be added to the object,” and accordingly enforces rigidity in regions far from contact. As a result, it reports cases such as “strong deformation of cloth” as failure modes.
>
> In contrast, our work specifically aims to address this limitation of prior approaches that “do not readily extend to more general, large-scale non-rigid deformations” (L48–49) by incorporating physics-based simulation. We would like to highlight that, for this target problem, we outperform all relevant baselines, as shown in Tab. R2 above. We will include this discussion in the revision.

---

### Author Response · Authors · 2025-11-25

We thank the reviewers and the AC for their time and effort in evaluating our paper. We have updated the manuscript by incorporating the suggested additional experiments (highlighted in blue in the PDF). In addition, we provide a supplementary video with further qualitative comparisons at this [anonymous link](https://youtu.be/0SzpruFEou4). Please let us know if you have any remaining concerns or questions—we will do our best to address them.

---

### Meta-Review · Area_Chair_zV4T · 2026-01-07

**Summary:**

The paper proposes PhysHandi, a physics-based framework for jointly reconstructing 3D hands and highly deformable objects from sparse-view RGB-D input using dense hand modeling and physical simulation.

The reviewers acknowledge that the paper tackles an important and underexplored problem, joint reconstruction of hands and highly deformable objects and appreciate the effort to integrate physics-based simulation with dense hand modeling.

However, the consensus is that the technical advance over prior work (notably PhysTwin) is perceived as incremental, and that the empirical evidence is not strong enough to clearly support the paper’s central claims.

In particular, concerns remain regarding novelty, strength of quantitative improvements, evaluation of hand reconstruction quality, and modeling assumptions. While the rebuttal added clarifications and additional experiments, these were not sufficient to fully address the core reservations. As a result, the paper does not meet the bar for acceptance at this time.

**Reviewer Concerns:**

Concerns partially addressed by the rebuttal:
- Clarification of the relationship to PhysTwin, including motivation for dense MANO-based hand modeling and inverse-physics refinement.
- Additional metrics (e.g., deformation-focused CD_dyn) and comparisons to strengthen the case that improvements are more visible in dynamically deforming regions.
- Added robustness experiments under noisy inputs and qualitative results via supplementary videos.
- Runtime analysis showing comparable or slightly improved efficiency relative to PhysTwin.

Remaining concerns:
- Novelty remains limited: multiple reviewers still view the method as an incremental extension of PhysTwin rather than a fundamentally new modeling or learning paradigm.
- Hand reconstruction claims are not fully convincing: direct, comprehensive evaluation of hand accuracy (especially beyond single-view or indirect metrics) is still seen as insufficient.
- Quantitative gains are modest on standard metrics, and improvements rely heavily on newly introduced or task-specific evaluation choices.
- Modeling assumptions, such as fixed hand–object contact topology, restrict applicability and remain a significant limitation.
- Some reviewers question whether the work is better characterized as object simulation with hand refinement, rather than a balanced hand–object reconstruction framework.

Overall, while the rebuttal improves clarity and presentation, it does not fundamentally change the reviewers’ assessment of contribution strength.

**Reviewer Scores:**

- Reviewer rNHX: Likely remains Reject primary concerns about limited hand evaluation, modest gains, and modeling assumptions persist, despite clarifications.
- Reviewer asRB: Likely keep 6, but still views the contribution as incremental.
- Other reviewers: no clear shift toward acceptance.

Overall assessment:
The rebuttal led to incremental improvements in clarity and evidence, but not enough to overcome the shared concerns regarding novelty and strength of contribution. The final decision remains Reject.

---

### Decision · Program_Chairs · 2026-01-26

Reject